# The advective Brewer-Dobson circulation in the ERA5 reanalysis: climatology, variability, and trends

Mohamadou Diallo[1], Manfred Ern[1], and Felix Ploeger[1,2]

[1]Institute of Energy and Climate Research, Stratosphere (IEK–7), Forschungszentrum Jülich, 52 425 Jülich, Germany.
[2]Institute for Atmospheric and Environmental Research, University of Wuppertal, Wuppertal, Germany.

**Correspondence:** Mohamadou Diallo (m.diallo@fz-juelich.de)

**Abstract.**

The stratospheric Brewer-Dobson circulation (BDC) is an important element of climate as it determines the transport and distributions of key radiatively active atmospheric trace gases, which affect the Earth's radiation budget and surface climate.

Here, we evaluate the inter-annual variability, climatology and trends of the BDC in the ERA5 reanalysis and inter-compare with its predecessor, the ERA-Interim reanalysis, for the 1979–2018 period. We also assess the modulation of the circulation by the Quasi-Biennial Oscillation (QBO) and the El Niño-Southern Oscillation (ENSO), and the forcings of the circulation by the planetary and gravity wave drag. The comparison of ERA5 and ERA-Interim reanalyses shows a very good agreement in the morphology of the BDC and in its structural modulations by the natural variability related to QBO and ENSO. Despite the good agreement in the spatial structure, there are substantial and significant differences in the strength of the BDC and natural variability impacts on the BDC between the two reanalyses, particularly in the upper troposphere and lower stratosphere (UTLS), and in the upper stratosphere. Throughout most regions of the stratosphere, the variability and trends of the advective BDC are stronger in the ERA5 reanalysis due to stronger planetary and gravity wave forcings, except in the UTLS below 20 km where the tropical upwelling is up to 40 % weaker mainly due to a significantly weaker gravity wave forcing at the equatorial-ward upper flank of the subtropical jet. In the extra-tropics, the large-scale downwelling is stronger in ERA5 than in ERA-Interim linked to significant differences in planetary and gravity wave forcings in the upper stratosphere. Analysis of the BDC trend shows a global insignificant acceleration of the annual mean residual circulation with an acceleration rate of about 1.5 % per decade at 70 hPa due to the long-term intensification in gravity and planetary wave breaking, consistent with observed and modelled BDC changes.

Our findings suggest that the advective BDC from the kinematic ERA5 reanalysis is well suited for climate model validation in the UTLS and mid-stratosphere when using the standard formula of zonally-averaged zonal momentum equation. The reported differences between the two reanalyses may also affect the nudged climate model simulations. Therefore, additional studies are needed to investigate whether or not nudging climate models toward ERA5 reanalysis will reproduce the upwelling trends from free-running simulations and from ERA5. Finally, further studies are also needed to better understand the impact of the new non-orographic gravity wave parameterization scheme, higher model top, and the representation of the sponge layer in ERA5 on the differences in the upper stratosphere and polar regions.

## 1 Introduction

As a key element of the Earth's climate system, the stratospheric Brewer-Dobson circulation (BDC, e.g. Brewer, 1949; Butchart, 2014) has received a lot of interest during the last decades because of its role in climate change and weather prediction
(WMO, 2018; Baldwin and Dunkerton, 2001). The BDC determines the transport and distribution of key stratospheric trace gases like ozone, water vapor, and aerosol, which affect the Earth's radiation budget and surface climate (Forster and Shine, 1997; Riese et al., 2012). Changes in the distribution of these trace gases most effectively impact surface climate, in particular, if they occur in the upper troposphere and lower stratosphere (UTLS) region (Lacis et al., 1998). Expected changes in the BDC as predicted by climate models will impact the trace gas composition in the UTLS and, thus, may have crucial consequences
for regional and global climate.

The BDC is defined as a slow wave-driven circulation in which air masses ascend in the tropics, drift poleward in the mid-latitude stratosphere (e.g. surf zone), and then are transported downward in the extratropical regions (Brewer, 1949; Holton et al., 1995; Butchart, 2014). This mean meridional transport of air masses can be characterized by the stratospheric mean mass flux as given by the *stratospheric residual circulation* and by two-way exchange due to eddy mixing (Waugh and Hall,
2002; Ray et al., 2010; Garny et al., 2014; Ploeger et al., 2015a; Miyazaki et al., 2016). The residual circulation represents an approximation of the diabatic or Lagrangian mean circulation as described in the Transformed Eulerian Mean framework (Andrews et al., 1987). The residual circulation can be separated into two branches: a deep branch driven by planetary waves and a shallow branch driven by synoptic and smaller-scale gravity waves (Plumb, 2002; Birner and Bönisch, 2011). In addition, Lin and Fu (2013) further separated the shallow branch into two sub-branches: the transition branch (at pressure 100-70 hPa) and
the shallow branch (at pressure 70-30 hPa). The two-way mixing is defined as a quasi-horizontal stirring and irreversible displacement of air masses induced by breaking of large- and small-scale waves in the surf zone, and turbulent mixing (McIntyre and Palmer, 1984; Randel, 1993; Shuckburgh and Haynes, 2003).

Driven by breaking waves in the stratosphere (Haynes et al., 1991; Rosenlof and Holton, 1993; Newman and Nash, 2000; Plumb, 2002; Shepherd, 2007) and varying on seasonal to decadal timescales with strongest downwelling over the winter pole
(Bönisch et al., 2011), the stratospheric residual circulation is further modulated by natural variability, including the El Niño–Southern Oscillation (ENSO) (Randel et al., 2009; Diallo et al., 2018, 2019), the Quasi-Biennial Oscillation (QBO) (Baldwin et al., 2001; Ern et al., 2014), volcanic aerosols (Thompson and Solomon, 2009; Garfinkel et al., 2017; Diallo et al., 2017), increasing Greenhouse Gas (GHG) levels (Butchart et al., 2010) and ozone depleting substances (Li et al., 2008; Polvani et al., 2018). The interannual variability of the residual circulation is mostly induced by two major modes of climate variability, the
QBO and the ENSO, which trigger a modulation of vertical transport in the stratosphere by affecting the temperature structure and thus the tropical upwelling and extratropical downwelling (Plumb and Bell, 1982; Trepte and Hitchman, 1992; Niwano et al., 2003; Punge et al., 2009). Future projections of climate models predict changes in the wave propagation due to increasing GHG concentration levels (Shepherd and McLandress, 2011), which, in turn, alters the stratospheric residual circulation and

its modulations by climate variability modes (Saravanan, 1990; van Oldenborgh et al., 2005; Latif and Keenlyside, 2009; Cai et al., 2014; Kawatani and Hamilton, 2013).

Climate models predict that increasing GHG levels will globally strengthen the BDC (e.g., Butchart et al., 2010; Garny et al., 2011), consistent with observed negative temperature trends in the tropical lower stratosphere (Thompson and Solomon, 2005; Fu et al., 2019). This acceleration of the BDC results from the upward shift of the critical layer of wave breaking due to the strengthening upper flank of the subtropical jet (Shepherd and McLandress, 2011). Reanalyses estimates of mean age of stratospheric air (i.e. the average transit time of air parcels through the stratosphere) show robust evidence for strengthening of the shallow branch and the southern hemispheric deep branch of the BDC (e.g., Bönisch et al., 2011; Diallo et al., 2012; Monge-Sanz et al., 2012; Chabrillat et al., 2018; Ploeger et al., 2019), consistent with BDC trends derived from trace gas observations, including $CO_2$, $SF_6$ and $N_2O$ (Stiller et al., 2012; Hegglin et al., 2014; Ray et al., 2010, 2014; Haenel et al., 2015). The southern deep branch is strongly modulated by the changes in ozone depletion (e.g. Lin and Fu, 2013; Polvani et al., 2018), therefore, its changes will be affected by the ozone recovery. These findings are also consistent with the advective BDC trends found in reanalyses by Abalos et al. (2015). An updated time series of BDC changes since 1976 derived from in-situ observations of $CO_2$ and $SF_6$ in the northern hemisphere middle stratosphere (Engel et al., 2009, 2017) shows no significant long-term BDC trend between 27 and 32 km, consistent with the mean age of air trends derived from the ERA-Interim and with some climate models (e.g., Garfinkel et al., 2017), but inconsistent with other known reanalyses (Chabrillat et al., 2018; Ploeger et al., 2019). Linz et al. (2017) reported that many climate models are not yet able to reproduce the strength of the BDC as determined from observations. The fact that there is no clear observational evidence of northern hemispheric deep branch changes challenges both the validity of climate model predictions and observational uncertainties in the northern hemisphere middle stratosphere. According to notable source of uncertainty in reanalyses and for future climate projections lies in the strength of the BDC changes. Notable source of uncertainty in reanalyses and in future climate projections lies in the trend and strength of the BDC changes (Chabrillat et al., 2018; Ploeger et al., 2019; Eichinger et al., 2019). Robust knowledge of the natural variability of the BDC on seasonal to decadal timescales, in turn, is a prerequisite for the reliable detection and attribution of long-term anthropogenically-forced trends.

Commonly used as a basis for comparisons of the predicted BDC changes in climate models and climate model nudging (Dietmüller et al., 2017; Polvani et al., 2018; Chrysanthou et al., 2019; Davis et al., 2020), the reanalysis data sets provide the best knowledge of the past and present atmospheric state by combining models with observations. Recently, the European Centre for Medium-Range Weather Forecasts (ECMWF) released its fifth generation of atmospheric reanalysis: the ERA5 global reanalysis (Hersbach et al., 2020). Built to replace the ERA-Interim reanalysis (Dee et al., 2011), this newly available high-resolution reanalysis is expected to be a milestone for meteorological analysis as it includes extensive improvements in the representation of atmospheric processes compared to the previous generations of reanalyses. Hence, it is of key importance to evaluate the representation and characteristics of the BDC in ERA5 reanalysis as well as its consistency with the ERA-Interim reanalysis.

In the present study, we seek to objectively evaluate the representation of the advective BDC in the ERA5 reanalysis in comparison with the ERA-Interim reanalysis, using different diagnostics, including the residual circulation transit time, the residual

vertical velocity, the residual mass stream-function and its modulation by the QBO and ENSO signals (Butchart, 2014; Abalos et al., 2015; Ploeger et al., 2015b, a). We describe the reanalysis data sets and methods used in this study, including the Transformed Eulerian Mean and the statistical approaches, in Section 2. The mean climatology of the upwelling/downwelling mass flux and the inter-annual variability from the ERA5 reanalysis are discussed and compared with the ERA-Interim reanalysis in Section 3.1. The residual stream-function and transit times are shown in Section 3.2. Section 3.3 presents the effects of different variability modes on the zonal mean wind, temperatures, residual vertical velocity and mass stream-function estimated using a statistical prediction model. Section 3.4 shows the modulation of the advective BDC by the planetary and gravity wave drag. Section 4 presents the trend in the advective BDC together with the trends in planetary and gravity wave drag. Finally, Section 5 provides further discussions and conclusions.

## 2  Data and Methodology

### 2.1  Description of the reanalyses

The wind and temperature data used in this study are from the ERA5 and the ERA-Interim reanalyses, provided by the ECMWF. The ERA5 reanalysis is based on the Integrated Forecasting System (IFS) and benefits from a decade of progress in model physics, dynamics, and data assimilation. Compared to ERA-Interim, ERA5 has better temporal resolution (6-hourly versus 1-hourly) and better horizontal (31 km versus 80 km) and vertical (60 levels versus 137 levels) resolutions and extends higher into the middle atmosphere (0.1 hPa versus 0.01 hPa or 65 km versus 80 km). The ERA5 reanalysis has replaced the ERA-Interim reanalysis from 31 August 2019. Due to stratospheric temperature biases for the 2000-2006 time period exhibited in the first version of ERA5, the ECMWF has published the ERA5.1 to improve upon the cold bias in the lower stratosphere seen in ERA5 (Simmons et al., 2020). In addition, a warm bias higher up above 40 km persists for much of the period from 1979 (Hoffmann et al., 2019). In addition to the higher spatial and temporal resolution, other key improvements of ERA5 compared to ERA-Interim are a better ability to resolve synoptic-scale features like hurricanes and tropical cyclones as well as a better representation of the tropospheric circulation. Moreover, data from many recent satellite instruments are now additionally assimilated (Li et al., 2020; Hoffmann et al., 2019). Potential limitations of reanalyses, including ERA5 and ERA-Interim, are non-physical trends and variability due to changes in the observing system such as the introduction of COSMIC radio occultation data in 2006, which affects the variability of temperatures near the tropical tropopause.

In this study, the wind and temperature fields from both reanalyses have been interpolated onto a $1° \times 1°$ longitude and latitude grid, and are extracted from the analysis available at 6 h interval on their original model levels interpolated to log-pressure levels for the residual circulation calculations. The difference between ERA-Interim and ERA5 is that the assimilation cycles start at 06:00 UT and 18:00 UTC for ERA5 and at 00:00 UTC and 12:00 UTC for ERA-Interim. In addition, the dynamical fields in ERA5 are archived hourly, and no longer as accumulation over every 3 hours as in ERA-Interim. Note that both reanalyses are part of the SPARC Reanalysis Intercomparison Project (S-RIP) (Fujiwara et al., 2017), which is a coordinated inter-comparison of modern global atmospheric reanalyses. In particular, the Chapter 5 of the S-RIP report evaluates several

commonly used metrics of the BDC calculated from reanalysis fields. This work contributes to this assessment of the BDC metrics in the ERA5 reanalysis.

## 2.2 Metric of advective Brewer-Dobson circulation

Commonly used as a proxy for the advective BDC, the Transformed Eulerian Mean (TEM) residual circulation is derived from the standard formula of zonally-averaged zonal momentum equation, in latitude and log-pressure coordinates $(\phi, z)$, which is given by Andrews et al. (1987)

$$\frac{\partial \overline{u}}{\partial t} + \overline{v} \cdot \left[ \frac{1}{a \cdot \cos \phi} \cdot \frac{\partial}{\partial \phi} (\overline{u} \cdot \cos \phi) - f \right] + \overline{w} \cdot \frac{\partial \overline{u}}{\partial z} = \overline{\mathfrak{I}} \tag{1}$$

where $a$ is the Earth's radius, $\phi$ is latitude. $z = -H \cdot \ln(\frac{p}{p_s})$ is the log-pressure height (vertical coordinate), $H$ is a constant height defined as the scale height $(= R \cdot T_s/g)$ for the log-pressure coordinate taken as $7\,km$, $T_s(= 240\,K)$ is chosen as standard reference temperature, $R$ is the gas constant for dry air, $p_s(= 1013\,hPa)$ is chosen as standard reference pressure, $(u, v, w)$ are respectively the zonal mean, meridional and vertical wind velocity. $\overline{\mathfrak{I}}$ corresponds to the total zonal momentum forcing. The Coriolis frequency is $f = 2 \cdot \Omega \cdot \sin \phi$ and $\Omega$ is the Earth's rotation rate. Here, and in the following, the zonal mean and the deviations from the zonal mean are respectively indicated by an over-bar and a prime. The partial derivatives with respect to $z$ (vertical) and $\phi$ (meridional) direction are indicated by $\frac{\partial}{\partial z}(.)$ and $\frac{\partial}{\partial \phi}(.)$ respectively.

As the advective BDC is a Lagrangian mean circulation, the Eulerian mean velocity is not a sufficient diagnostic metric. However, a useful proxy for the Lagrangian mean circulation under time-averaged conditions is provided by the residual mean meridional circulation $(\overline{v^*}, \overline{w^*})$ in the TEM framework (Andrews and McIntyre, 1976, 1978; Dunkerton et al., 1978; Holton, 1990). The latitudinal and vertical components of the residual mean meridional circulation $(\overline{v^*}, \overline{w^*})$ and the mean mass streamfunction of the residual circulation, $\psi^*(\phi, z)$, are given as follows

$$\overline{v^*} = \overline{v} - \frac{1}{\rho_{sc}} \cdot \left( \frac{\rho_{sc} \cdot \overline{v' \cdot \theta'}}{\overline{\theta}_z} \right)_z = -\frac{1}{\rho_{sc} \cdot \cos \phi} \cdot \frac{\partial \psi^*}{\partial z} \tag{2}$$

$$\overline{w^*} = \overline{w} + \frac{1}{a \cos \phi} \cdot \left( \cos \phi \cdot \frac{\overline{v' \cdot \theta'}}{\overline{\theta}_z} \right)_\phi = \frac{1}{a \cdot \rho_{sc} \cdot \cos \phi} \cdot \frac{\partial \psi^*}{\partial \phi} \tag{3}$$

Introducing the equations (2, 3) into the equation (1) will lead to the TEM momentum equation (Andrews and McIntyre, 1978)

$$\frac{\partial \overline{u}}{\partial t} + \overline{v^*} \cdot \left[ \frac{1}{a \cdot \cos \phi} \cdot \frac{\partial}{\partial \phi} (\overline{u} \cdot \cos \phi) - f \right] + \overline{w^*} \cdot \frac{\partial \overline{u}}{\partial z} = \overline{X}_{urGWD} + DF = \overline{\mathfrak{I}} \tag{4}$$

where $\overline{X}_{urGWD}$ is the residual mean nonconservative forcing unresolved by the model and $DF$ is the normalised EP-flux divergence. The $\overline{X}_{urGWD}$ contribution consists of parameterized gravity wave drag and further imbalances of the reanalysis momentum budget that are caused by the data assimilation system, and that can be interpreted as another contribution of

unresolved gravity waves (e.g., Alexander and Rosenlof, 1996; McLandress et al., 2012; Ern et al., 2014). The density $\rho = \rho_o \cdot exp(-z/H)$ is the stratification of the atmosphere. $\theta$ is the potential temperature (hence, $w = -(H/p) \cdot (dp/dt)$, where $p = p_o \cdot exp(-z/H)$ is the pressure). The Eliassen-Palm (EP) flux, $\mathbf{F}$, decreases exponentially with height due to the scaling factor $\rho_{sc} = \exp(-z/H) = \rho/\rho_o$. $DF$ can be decomposed into resolved planetary ($\overline{X}_{PWD}$) and resolved gravity ($\overline{X}_{GWD}$) wave drags and is written as following

$$DF = \frac{\nabla \cdot \mathbf{F}}{a \cdot \rho_{sc} \cdot \cos\phi} = \frac{1}{a \cdot \cos\phi} \cdot \frac{\partial}{\partial\phi}(F_\phi \cdot \cos\phi) + \frac{\partial F_z}{\partial z} = \overline{X}_{PWD} + \overline{X}_{GWD} \tag{5}$$

with $\mathbf{F} = \{F_\phi, F_z\}$ the EP-flux vector, with respective latitudinal and vertical components

$$F_\phi = a \cdot \rho_{sc} \cdot \cos\phi \cdot \left( \overline{u}_z \cdot \frac{\overline{v' \cdot \theta'}}{\overline{\theta}_z} - \overline{v' \cdot u'} \right) \tag{6}$$

$$F_z = a \cdot \rho_{sc} \cdot \cos\phi \cdot \left\{ \left[ 2 \cdot \Omega \cdot \sin\phi - \frac{1}{a \cdot \cos\phi} \cdot \frac{\partial}{\partial\phi}(\overline{u} \cdot \cos\phi) \right] \cdot \frac{\overline{v' \cdot \theta'}}{\overline{\theta}_z} - \overline{w' \cdot u'} \right\} \tag{7}$$

$DF$ represents an important part of the forcings of the mean mass flux circulation from the dissipation of resolved planetary and gravity waves. $DF$ can be decomposed into wave numbers ranging from 1 to 180 following previous studies (Ern et al., 2014; Alexander and Rosenlof, 1996). The resolved planetary wave drag, $\overline{X}_{PWD}$, is estimated as the integration over zonal wave numbers ranging between 1–20 in equation (5). The total gravity wave drag, $\overline{X}_{GWD}$, is the sum of the small contribution of explicitly resolved gravity waves with zonal wave numbers larger than 20 in equation (5), parameterized gravity wave drag and the momentum budget imbalances induced by data assimilation. In the reanalyses, the missing wave drag can be considered equal to the unresolved wave drag by the ECMWF model grid, therefore, contributing to the gravity wave forcings in the momentum equation (eq. 4). Thus, the sum of this missing wave drag and the integrated model-resolved wave drag between the 21 and 180 zonal wave numbers gives the estimate of the total gravity wave drag (Alexander and Rosenlof, 1996). For additional details please see Ern et al. (2014, 2015, 2016).

If we consider that $\overline{X}_{urGW} \neq 0$ and $DF \neq 0$, representing the zonal forcing due the wave activity such as planetary and gravity waves, then, to maintain the steady conditions, the BDC has to stay non-zero (Andrews and McIntyre, 1976, 1978; Eliassen and Palm, 1961; Charney and Drazin, 1961). Hence under this approximation, the circulation resulting from the wave forcings is the BDC. Further, the mass stream-function of the residual circulation, $\psi^*(\phi,z)$, is obtained by integrating vertically the total forcing term from the equation (4). Using the downward control principle (e.g. Haynes et al., 1991; Rosenlof and Holton, 1993; Garcia and Boville, 1994), it can be written as

$$\psi^*(\phi,z) = \int_z^\infty \left\{ \frac{a^2 \cdot \rho_{sc} \cdot \cos^2\phi}{\overline{m}_\phi} \left( DF + \overline{X}_{urGW} - \frac{\partial\overline{u}}{\partial t} \right) \right\}_{\phi=\phi(z')} dz' \tag{8}$$

where the angular momentum per unit mass is defined by $\overline{m}(\phi) = a \cdot \cos\phi \cdot (\overline{u} + a \cdot \Omega \cdot \cos\phi)$. At a line, where angular momentum is constant, the mass stream-function is integrated along a contour $\phi(z)$.

The tropical upwelling mass flux $\mathscr{F}$ is given by

$$\mathscr{F} = 2\pi \cdot a^2 \int_{\phi_-}^{\phi_+} \rho_{sc} \cdot \overline{w^*} \cdot \cos\phi \cdot d\phi \qquad (9)$$

This upward tropical mean mass flux, that is the mass per unit time, is integrated for the whole set of latitude circles where the vertical residual component is positive (i.e. $\overline{w^*} > 0$).

In addition to the mean mass flux and the residual mean mass stream-function diagnostics, we also calculated the residual circulation transit time (RCTT) using a two dimensional backward trajectory model driven by the residual mean meridional circulation ($\overline{v^*}$, $\overline{w^*}$). Used as a metric of the advective BDC strength, the RCTT is defined as transit time of an air parcel that is only advected by the residual circulation through the stratosphere. For additional details about the method of the RCTT estimates see Ploeger et al. (2015b).

## 2.3 Statistical prediction model

For appropriately quantifying the QBO- and ENSO-induced variability in the advective BDC, different metrics of the circulation, including the temperature, zonal wind, $\overline{w^*}$ and $\overline{\psi^*}$ are analyzed using an established statistical prediction model. This regression analysis model has been described in details and applied in our previous studies (Diallo et al., 2012, 2017, 2018, 2019; Tao et al., 2019). The statistical model is based on the principle that the monthly zonal mean time-series of the BDC metrics can be decomposed into a sum of different contributions, including a linear trend, QBO, ENSO, seasonal cycle and a residual ($\varepsilon$). For a given metric of the advective BDC strength, $BDC_{metric}$ (herein U, T, $\overline{w^*}$, $\psi^*$ and wave drags), at any given latitude ($\phi$) and altitude (z) position in the stratosphere, the regression model can be written simply as following

$$BDC_{metric}(t,\phi,z) = Trend(t,\phi,z) + SeasCycle(t,\phi,z) + QBO(t - \tau_{qbo}(\phi,z),\phi,z) + ENSO(t - \tau_{enso}(\phi,z),\phi,z) + \varepsilon(t,\phi,z) \quad (10)$$

The estimated QBO and ENSO coefficients with the regression fit as a function of latitude and altitude are normalized by the standard deviation (SD) of the QBO and ENSO predictors, which are the QBO index at $50\,hPa$ and the Multivariate ENSO Index (Wolter and Timlin, 2011). We named these normalized coefficients as the *QBO* and *ENSO amplitude variations*. Because of the presence of lags ($\tau_{qbo}$ and $\tau_{enso}$,) in the QBO and ENSO terms, the problem is nonlinear and the residual may have multiple minima as a function of the parameters. In order to determine the optimal values of the lags the residual is first minimized at fixed lag and then selected from a range of possible lags. The statistical prediction model estimates an uncertainty using a Student's t-test (Zwiers and von Storch, 1995; Friston et al., 2007). Note that the contributions of volcanic aerosol and solar cycle are intentionally omitted for simplifying the description as these terms are not diagnosed in this study. For more details, please see our previous studies (Diallo et al., 2012, 2017, 2018, 2019; Tao et al., 2019).

# 3 Climatological advective BDC and its modulations

## 3.1 Inter-annual variability of the upwelling

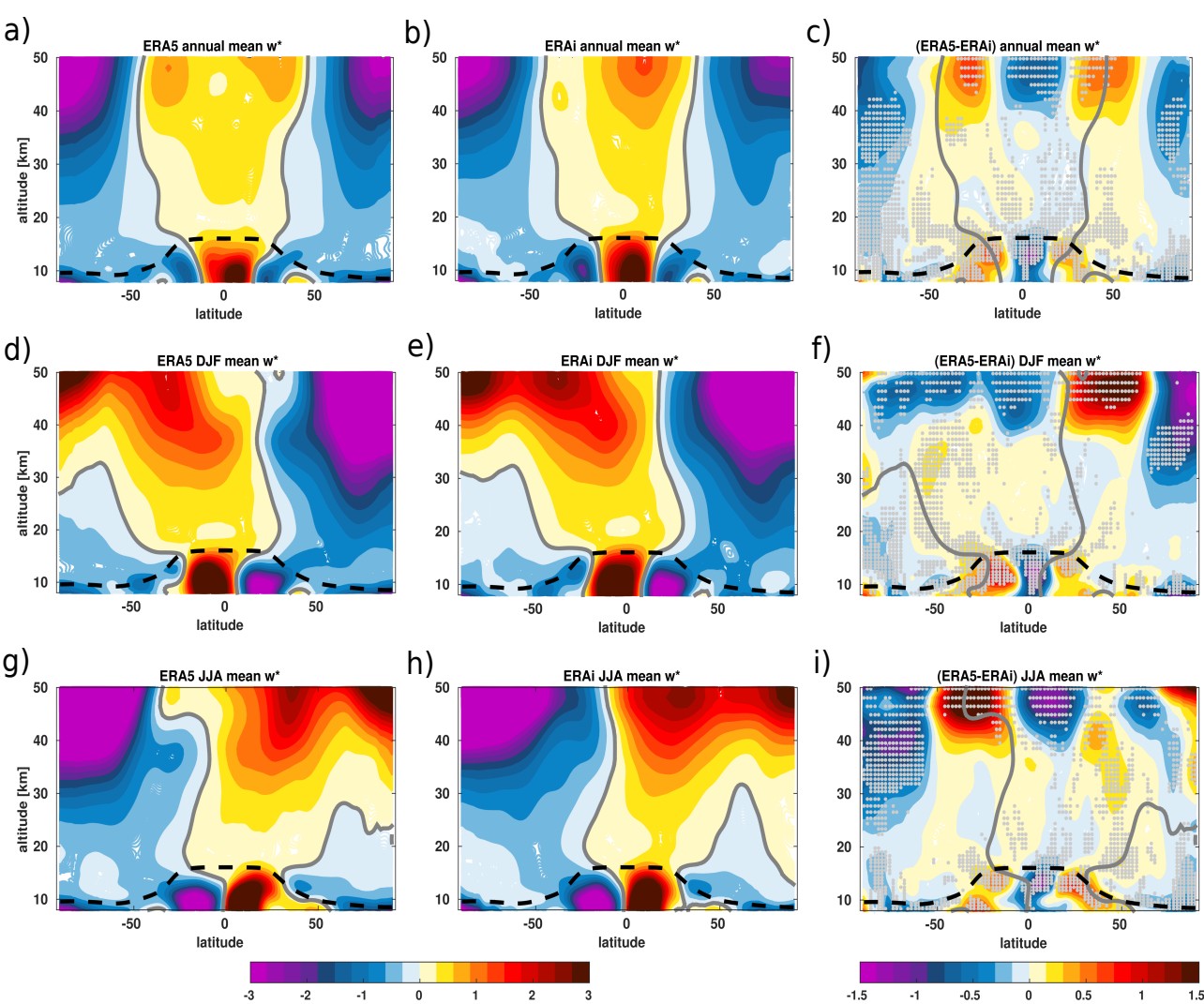

**Figure 1.** Annual (**a**, **b**), DJF (**d**, **e**) and JJA (**g**, **h**) mean variations of the residual vertical velocity ($\overline{w^*}$ in mm $\cdot$ s$^{-1}$) from the ERA5 (left column) and the ERA-Interim (middle column) reanalyses together with the associated differences (**c**, **f**, **i**) between the ERA5 and the ERA-Interim reanalyses (right column) for the 1979–2018 time period. Grey line indicates the zero $\overline{w^*}$ contours. Grey dots in panels (**c**, **f**, **i**) indicate regions where the differences between the ERA5 and ERA-Interim reanalyses are statistically significant at 95% estimated using student t-test. The climatological tropopause from the ERA-Interim is indicated as the black dashed horizontal line.

Evaluating the large-scale ascent and descent of air masses is a prerequisite for disclosing possible biases in the morphology and strength of the advective BDC. Therefore, we compare the tropical upwelling and extratropical downwelling of the air mass flux from the ERA5 reanalysis to the ERA-Interim reanalysis using the residual vertical velocity $(\overline{w^*})$. Figure 1a–i show the annual mean and seasonal variations of the $\overline{w^*}$ estimated from the two reanalyses, together with the associated differences for the 1979-2018 time period. Overall, there is a remarkably good agreement between the two reanalyses in the main structure of tropical upwelling and extratropical downwelling derived from the annual and seasonal mean $\overline{w^*}$ climatology (Fig. 1a, b).

However, substantial and statistically significant differences at 95% confidence interval in the strength of the tropical up-welling and extratropical downwelling arise in three distinct regions of the stratosphere(tropical pipe, mid-latitude surf zone and polar regions) (Fig. 1c). In the tropical pipe region associated with large-scale upwelling (e.g., Neu and Plumb, 1999; Ray et al., 2014), the ERA5 reanalysis exhibits a statistically significant weaker annual mean tropical upwelling in the UTLS, consistent with the well-known too fast ERA-Interim tropical diabatic upwelling compared to observations (Dee et al., 2011; Seviour et al., 2011; Ploeger et al., 2012). Using trace gas reconstruction and decomposition with a Lagrangian model driven by ERA-Interim, Ploeger et al. (2012) found tropical diabatic upwelling to be about 40% too fast in the tropical tropopause layer (Fueglistaler et al., 2009). The reduced tropical upwelling in ERA5 vertical residual circulation velocities, as compared to ERA-Interim, is also consistent with the diabatic ERA5 heating rates, which are 30-40% weaker in ERA5, therefore, correcting the known ERA-Interim bias (Ploeger et al., 2021). In the mid-latitude surf zone region associated with strong large-scale stirring, and poleward and downward transport (e.g., McIntyre and Palmer, 1984), the ERA5 reanalysis also shows a significant weaker downwelling. Conversely in the polar vortex region, the ERA5 reanalysis shows a stronger large-scale downwelling of the air mass than the ERA-Interim, suggesting potential differences in the polar vortex strength, but further studies are needed to prove that speculation. The maximum difference in the annual mean extratropical downwelling is about $0.5\,\mathrm{mm} \cdot \mathrm{s}^{-1}$ and occurs in the southern hemisphere. The residual velocity also exhibits a significant seasonal variation in these three distinct regions of the stratosphere with even larger differences between the two reanalyses in the upper stratosphere above $30\,\mathrm{km}$ (Fig. 1d–i). As the strength of advective BDC varies seasonally between the hemispheres, these large and significant differences in the $\overline{w^*}$ are displaced toward the winter hemisphere in the upper stratosphere with a maximum of about $1.5\,\mathrm{mm} \cdot \mathrm{s}^{-1}$. The reasons of these discrepancies in tropical upwelling and extratropical downwelling are likely associated with the differences in wave forcings resulting from the improvements in the ERA5, including higher vertical and horizontal resolutions, higher model top, and new gravity wave parameterization scheme (Orr et al., 2010; Hersbach et al., 2020).

To quantify the circulation differences between the two reanalyses, we average the annual mean $\overline{w^*}$ into vertical profiles based on the turnaround latitudes of tropical upwelling and extratropical downwelling together with the associated differences and their statistical significance at $2\sigma$ level for the 1979–2018 period (Fig. 2 a–f). The $\overline{w^*}$ vertical profiles show a good agreement in the $\overline{w^*}$ structure between the two reanalyses for the 1979-2018 time period (Fig. 2a–c). Despite the similarities in the vertical structure, the relative $\overline{w^*}$ differences show a significantly stronger large-scale downwelling in ERA5 than in ERA-Interim with a hemispheric asymmetry (Fig. 2 d–f). In the southern hemisphere, the large-scale downwelling differences are about $20\,\%$ stronger in ERA5 than in ERA-Interim below $30\,\mathrm{km}$, and are even larger above with a maximum of about $60\,\%$ at $35\,\mathrm{km}$ (Fig. 2 c, f). These differences are statistically significant at $2\sigma$ level as indicated by the errorbars. In the tropics, the

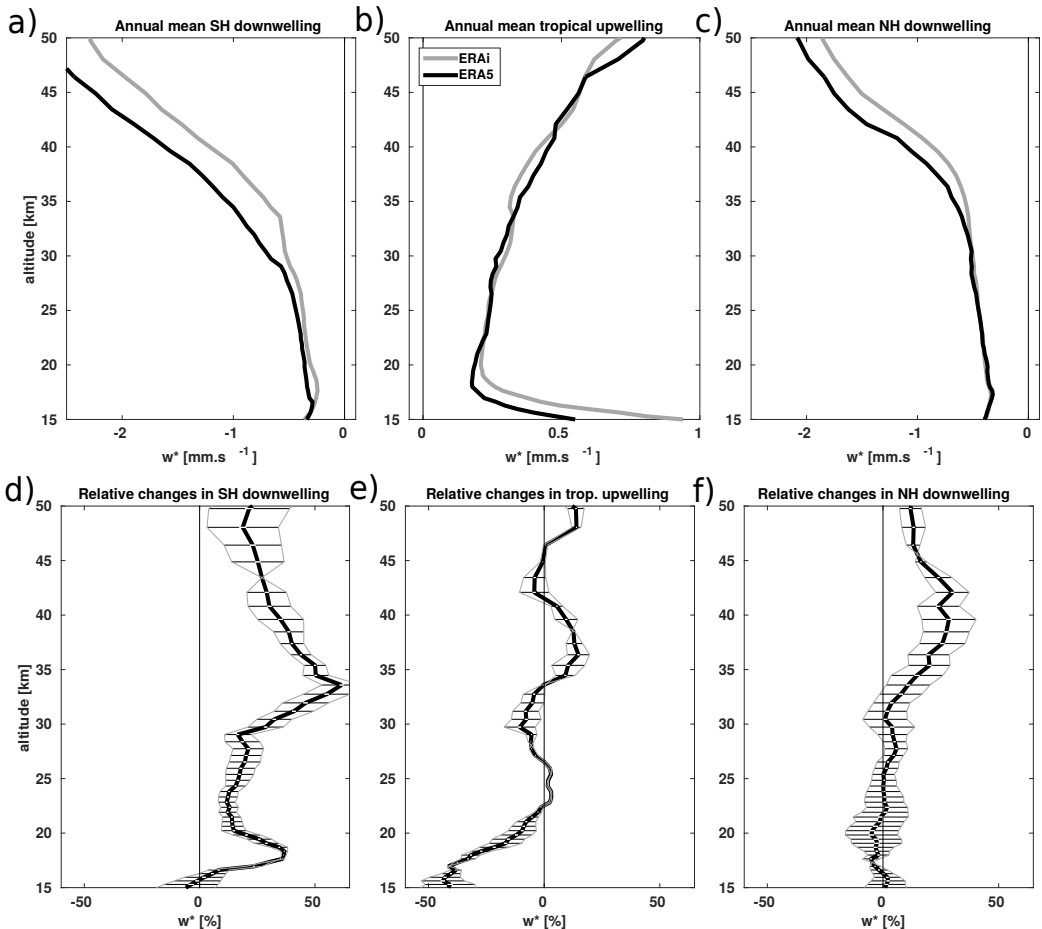

**Figure 2.** Vertical profiles of the turnaround annual mean $\overline{w^*}$ (in mm $\cdot$ s$^{-1}$): (**a**) SH downwelling, (**b**) tropical upwelling and (**c**) NH downwelling from the ERA5 reanalysis (black) and the ERA-Interim reanalysis (light-grey) for the 1979–2018 time period. Panels (**d**–**f**) show the ERA5 minus ERA-Interim $\overline{w^*}$, expressed as a percent difference relative to monthly mean. The error bars represent the $2\sigma$ significance level. Differences between the ERA5 and ERA-Interim reanalyses are significant where deviations from zero exceed the $2\sigma$ range.

differences in the large-scale upwelling is about 40 % weaker in ERA5 than in ERA-Interim below 17 km, and decreases to about 20-25 % at an altitude of 20 km (Fig. 2b, e). These differences are also statistically significant at $2\sigma$ and consistent with the stronger ERA-Interim upwelling in Fig. 1. Between 20 km and 35 km the differences are not statistically significant. Above 35 km, ERA5 reveals a statistically significant and stronger tropical upwelling than ERA-Interim. In the northern hemisphere, the downwelling differences between the two reanalyses are negligible below 30 km and reach 30 % above (Fig. 2c, f). The errorbars, which correspond to the statistically significant area at $2\sigma$ level, indicate that the large differences in upwelling and downwelling correspond to the regions where the variability is large. The observed differences in the upwelling and downwelling are likely due to a combination of several factors, including a higher resolution, higher model top, new non-

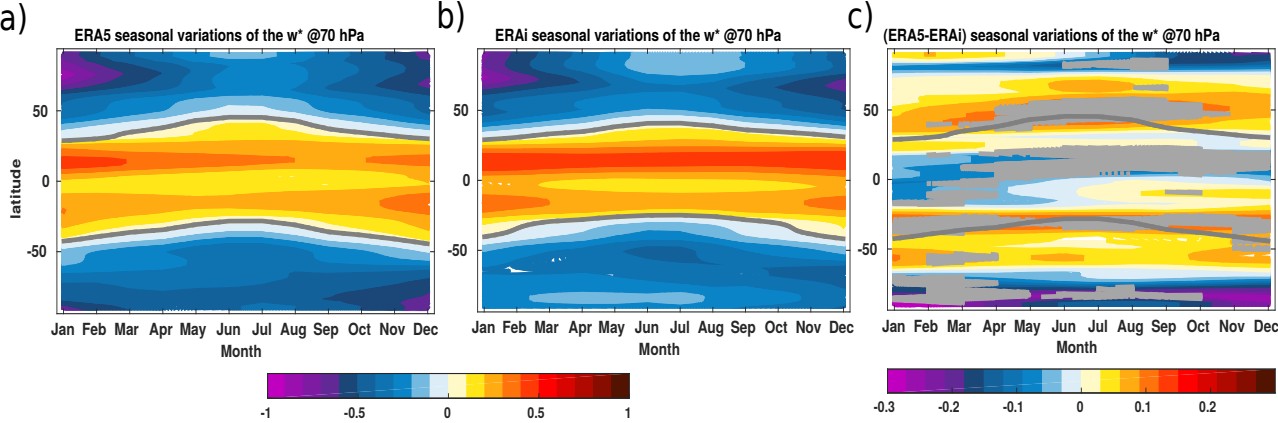

**Figure 3.** Monthly and latitudinal variations of monthly mean $\overline{w^*}$ (in mm · s$^{-1}$) at 70 hPa from the ERA5 reanalysis (**a**), the ERA-Interim reanalysis (**b**), and the difference between the ERA5 and the ERA-Interim reanalyses (**c**) for the 1979–2018 period. Grey horizontal lines indicate the zero $\overline{w^*}$ contours. Grey shading in panel (**c**) indicate regions where the differences between the ERA5 and ERA-Interim reanalyses are statistically significant at 95% confident level estimated using student t-test.

orographic gravity wave parameterization scheme, and the progress made in the more realistic representation of the sponge layer in ERA5 and will be discussed later (Sect. 3.4).

The seasonal variations in the $\overline{w^*}$ and its associated differences between the two reanalyses are also assessed at 70 hPa for the 1979-2018 time period (Fig. 3a–c). The 70 hPa pressure level is commonly used as the reference level for model inter-

comparisons of the upwelling strength of the air mass entering the stratosphere (e.g., Butchart et al., 2010). The tropical upwelling and extratropical downwelling patterns also agree well between the two reanalyses even in the variations of the upwelling zero-line. The $\overline{w^*}$ exhibits a 6–month phase shift between the northern and the southern hemisphere mid-latitudes, resulting from the correlation of the tropical upwelling annual cycle with the strongest descent located in the winter hemisphere (Fig. 3a, b). At 70 hPa, a significant seasonal variation of the $\overline{w^*}$ differences between the two reanalyses is also shown in

Figure 3c. The significantly weaker tropical upwelling in ERA5 varies seasonally between $0°$ and $20°$ N with a maximum of about $0.2$ mm · s$^{-1}$. In the mid-latitude, the large-scale downwelling is also significantly weaker and varies seasonally with a maximum occurring during the boreal summer (June-July-August). In the polar vortex region, ERA5 shows a significant large-scale downwelling, which is stronger than in ERA-Interim, as already evident from Fig. 1–2. The mass flux differences maximize in the polar vortex region of the southern hemisphere. These significant seasonal variations in the tropical upwelling

and the extratropical downwelling result from the pumping action of the wave forcings (Dunkerton et al., 1981; Randel, 1993; Holton et al., 1995; Plumb and Eluszkiewicz, 1999).

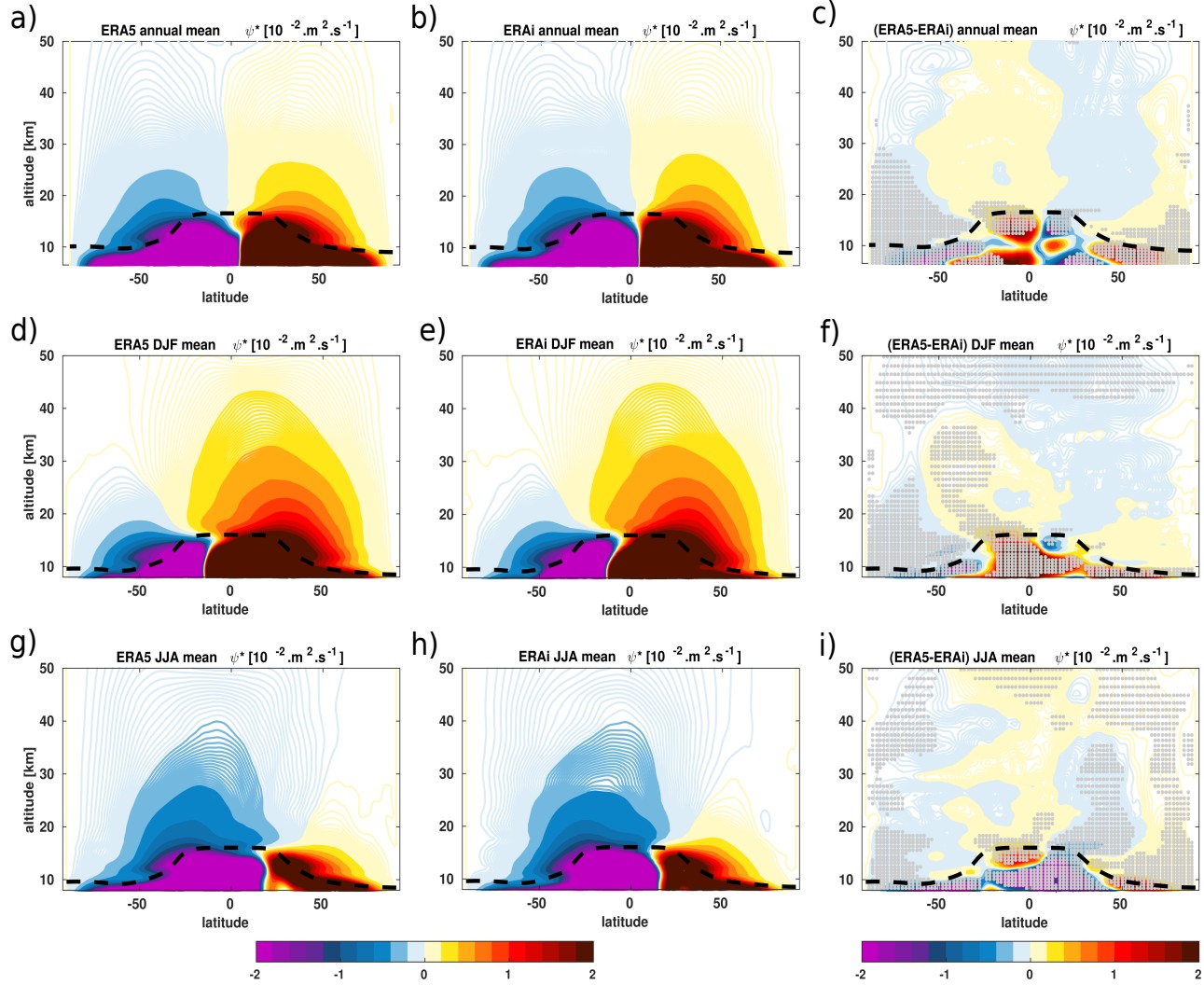

**Figure 4.** Zonal mean distribution of the annual (**a**, **b**), DJF (**d**, **e**) and JJA (**g**, **h**) mean variations of the residual stream-function ($\psi^*$ in $m^2 \cdot s^{-1}$) from the ERA5 (left column) and the ERA-Interim (middle column) reanalyses together with the associated differences (**c**, **f**, **i**) between the ERA5 and the ERA-Interim reanalyses (right column) for the 1979–2018 time period. The contours in the left and middle panels are spaced of about 0.015 $m^2 \cdot s^{-1}$ and the contours in the right panel are spaced of about 0.005 $m^2 \cdot s^{-1}$. Grey dots in panels (**c**, **f**, **i**) indicate regions where the differences between the ERA5 and ERA-Interim reanalyses are statistically significant at 95% estimated using student t-test. The climatological tropopause from the ERA-Interim is indicated as the black dashed horizontal line.

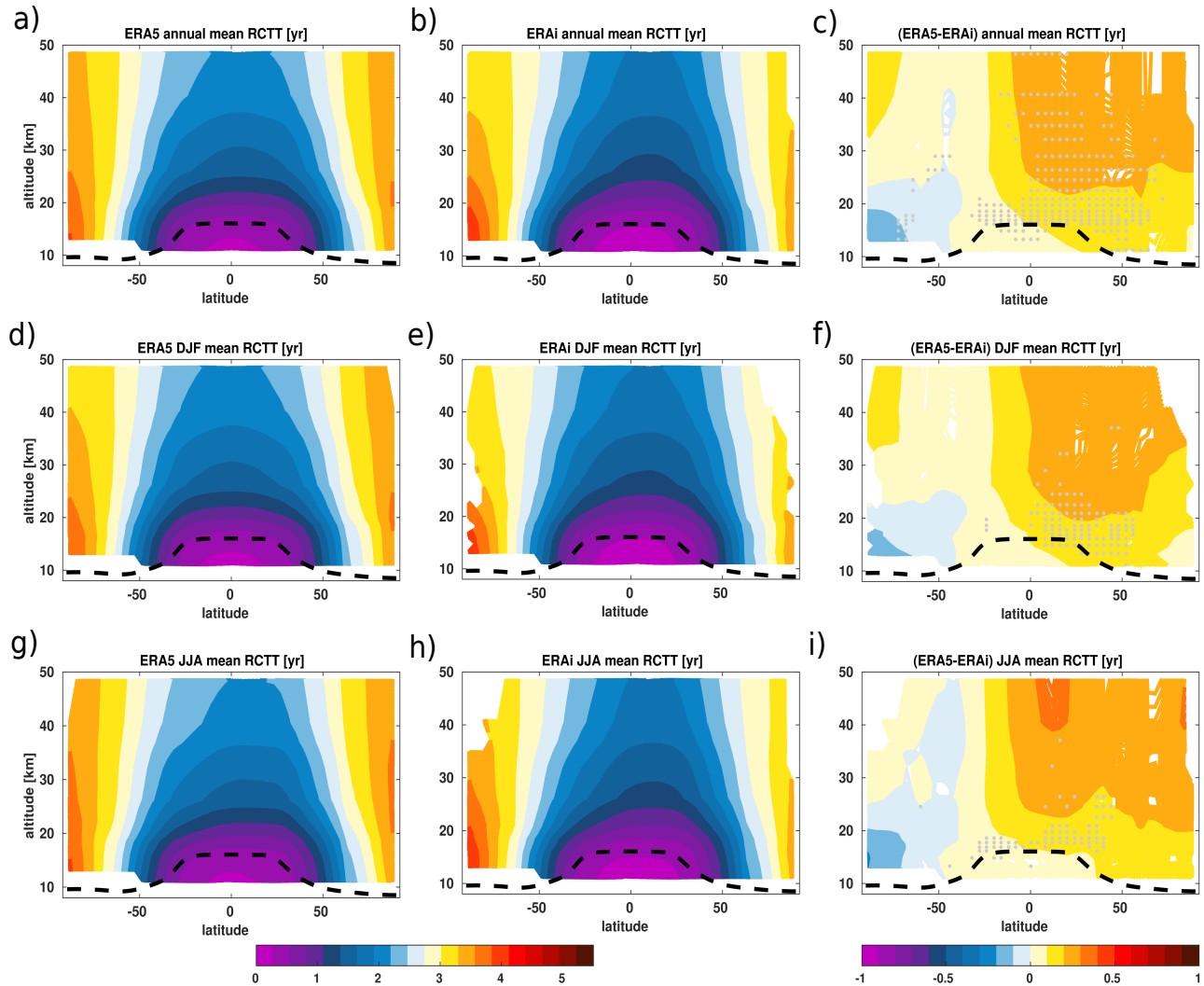

**Figure 5.** Zonal mean distribution of the annual (**a**, **b**), DJF (**d**, **e**) and JJA (**g**, **h**) mean variations of the the residual circulation transit time (RCTT in yr) from the ERA5 (left column) and the ERA-Interim (middle column) reanalyses together with the associated differences (**c**, **f**, **i**) between the ERA5 and the ERA-Interim reanalyses (right column) for the 1979–2018 time period. Grey dots in panels (**c**, **f**, **i**) indicate regions where the differences between the ERA5 and ERA-Interim reanalyses are statistically significant at 95% estimated using student t-test. The climatological tropopause from the ERA-Interim is indicated as the black dashed horizontal line.

## 3.2 Stratospheric residual circulation

In addition to the $\overline{w^*}$ analyses, we also evaluate the consistency and uncertainty in the residual circulation mass stream-function between the two reanalyses for the 1979-2018 time period. Figure 4a–i show a good agreement between the two reanalyses

in the morphology of the advective BDC, regarding an ascent of air mass in the tropics, a motion in the stratosphere toward the mid- and high altitudes and latitudes, and descent into the mid- and high latitude regions. Note that clear differences exist in the annual mean upwelling and downwelling circulation cells of the mass stream-function in the tropical pipe, mid-latitude surf zone, and polar vortex regions. Similarly to the $\overline{w^*}$, the residual circulation mass stream-function upwelling in the tropical

pipe and the mid-latitude surf zone is also weaker in ERA5 than in ERA-Interim, but stronger in the polar regions (Fig. 4a–c). The maximum and significant difference in the residual circulation also occurs in the UTLS below 20 km, consistent with the $\overline{w^*}$ differences. The seasonal variations in the residual circulation also agree well in the structure, but not the strength during the winter and summer between the two reanalyses (Fig. 4d–i). During the boreal winter (December-January-February), the differences in the mass stream-function between the two reanalyses are negative between 0 and $20° N$, which extends from

the UTLS into the upper stratosphere. This negative difference indicates a weaker circulation in ERA5 and is consistent with the weaker upwelling found in Fig. 3. However, the differences in the mass stream-function between the two reanalyses are significantly positive between $50° S$ and 0 and extend toward the southern hemisphere, leading to a stronger extratropical descent of the residual circulation in ERA5 than in ERA-Interim (Fig. 4d–f). The residual circulation is significantly slower in ERA5 than in ERA-Interim. During the boreal summer (June-July-August), these discrepancies in the residual circulation

are reversed (Fig. 4g–i)). These dominating seasonal features of the differences in the residual circulation suggest that the significant improvements in ERA5 likely induce a southward shift of upwelling cells in the UTLS during winter and northward shift during summer compared to ERA-Interim.

For further insights into the circulation differences between the two reanalyses, we also evaluate the annual and seasonal mean variations of the residual circulation transit time (RCTT), which is defined as the integrated time-scale of air mass

transport by the pure residual circulation and calculated using $\overline{v^*}$ and $\overline{w^*}$ in the equation (2) (e.g., Birner and Bönisch, 2011; Garny et al., 2014; Ploeger et al., 2015a, b). Note that the RCTT represents the integrated residual circulation effect, whereas the $\overline{w^*}$ is a local quantity. The RCTT shows a very good agreement in the morphology of the circulation between the ERA5 and ERA-Interim reanalyses, e.g. the shorter transit time in the tropics (faster BDC) and longer transit time in the extra-tropics (slower BDC) for the 2010-2018 period (Fig. 5a, b). However, the RCTT also shows clear and significant differences

in the residual circulation, consistent with the discrepancies in residual mass stream-function and residual vertical velocity previously discussed. The annual mean RCTT in ERA5 exhibits longer transit time (below about 0.5 years) associated with the significantly slower tropical ascent of the transition and shallow branch of the BDC (Lin and Fu, 2013; Diallo et al., 2019). The slow integrated residual circulation in the ERA5 reanalysis is consistent with the $\overline{w^*}$ differences as well as with the diabatic RCTT in ERA5 (Ploeger et al., 2021). In the northern hemisphere, the RCTT from the ERA5 reanalysis shows a longer transit

time (differences below about 0.5 years). In the southern hemisphere, the residence time in ERA5 tends to be longer than in ERA-Interim. The analysis of the seasonal variations also shows significant patterns of differences consistent with those shown in the annual mean RCTT (Fig. 5d–i). The differences vary seasonally and the maximum difference is found in the polar northern hemisphere near the polar vortex (Fig. 5f, i).

## 3.3 Natural variability related to QBO and ENSO

To further understand the linkage of differences between the two reanalyses with the impact of natural modes of climate variability, we analyze the representation of the QBO and ENSO variability. One of the major modes of variability in the ascending branch of the BDC on seasonal to intrannual timescales is the QBO (Lindzen and Holton, 1968; Plumb and Bell, 1982). Composed of alternating westerly and easterly zonal wind shears, the QBO propagates downward from the tropical middle stratosphere into the troposphere with a period of $\sim 28$ months. Both reanalyses agree well in the downward propagating QBO phases (Fig. 6a, b). The depiction of the QBO westerly and easterly phases from the lower to the upper stratosphere (from about 15 to $50\,km$) in ERA5 is very similar to the ERA-Interim for the 1979–2018 time period. The QBO disruption in January 2016, which was associated with the development of an easterly phase in the center of the westerly phase (Osprey et al., 2016; Newman et al., 2016), is clearly visible in both reanalyses. Apparent differences are also observed in the equatorial transitions in the eastward and westward zonal mean zonal wind and in the strength of QBO westerly and easterly phases between the ERA5 and ERA-Interim reanalyses. ERA5 reanalysis exhibits stronger QBO westerly and easterly phases compared to the ERA-Interim reanalysis in the tropical stratosphere above $20\,km$ (Fig. 6c). However, the QBO phases in the ERA5 reanalysis are weaker than in the ERA-Interim reanalysis between the 15 and $20\,km$. According to previous findings, the QBO westerly phase is sensitive to many model details, including the parametrized non-orographic gravity wave drag, and the vertical and horizontal resolution (Anstey et al., 2016; Geller et al., 2016; Polichtchouk et al., 2017, 2018; Shepherd et al., 2018). The amplitude of the westerly phase of the QBO and its frequency increase with enhanced non-orographic gravity waves. Therefore, the use of a new non-orographic gravity wave drag parameterization in ERA5 likely is the cause of the observed differences (Orr et al., 2010; Scaife et al., 2002; Lott et al., 2012; Richter et al., 2014). In ERA-Interim, Rayleigh drag was applied as a substitute for the non-orographic gravity wave drag. For ERA5, a Warner and McIntyre (2001) type non-orographic spectral gravity wave scheme was introduced as parameterization model and hence the Rayleigh drag could be switched off. In addition, the finer vertical and horizontal resolution in ERA5 likely leads to better representation of the synoptic and small-scale waves as sources of gravity waves are better represented, such as convection, which is important for gravity wave driving of the QBO (Lindzen and Fox-Rabinovitz, 1989).

Figure 7a–d shows the induced-QBO westerly and easterly amplitude variations in monthly mean zonal mean wind and temperatures estimated by fitting onto the QBO proxy with the statistical model (eq. 10) for the 1979–2018 time period. Both reanalyses agree very well in phase and periodicity of the westerlies and easterlies. In addition to gravity waves, the QBO is partly driven by waves trapped in the equatorial region (Wallace et al., 1993; Baldwin et al., 2001; Ern and Preusse, 2009; Ern et al., 2014). The QBO modulates the stratospheric residual circulation in the meridional and vertical direction by affecting temperature and tropical upwelling (Plumb and Bell, 1982; Collimore et al., 2003; Niwano et al., 2003; Punge et al., 2009). The QBO westerly shear is partly induced by equatorially trapped Rossby-gravity waves and the QBO easterly shear is induced by Kelvin waves (Holton and Lindzen, 1972; Plumb, 1977). Figure 7a, b show that the QBO westerly amplitude variation occurs between 15 and $25\,km$, the QBO easterly amplitude variation between 25 and $35\,km$ and again QBO westerly amplitude variation above $35\,km$. Note that the region of tropical westerly shear between 15 and $20\,km$ is associated with

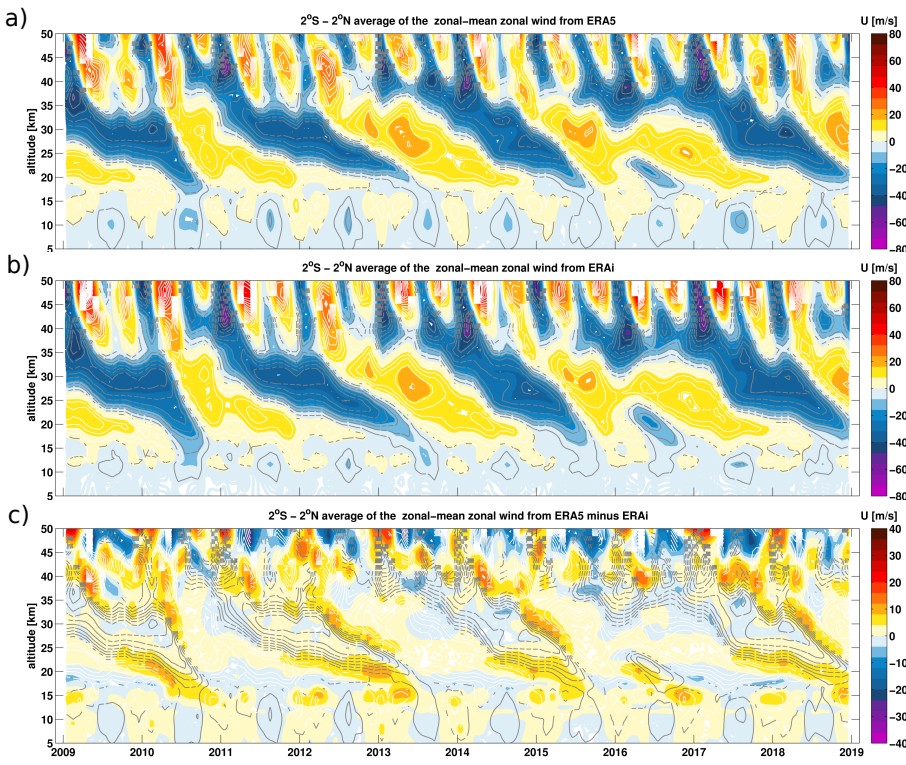

**Figure 6.** Tropical ($2^oS$ to $2^oN$ average) zonal-mean zonal wind (in m · s$^{-1}$) from the ERA5 reanalysis (**a**), the ERA-Interim reanalysis (**b**) and the difference between the ERA5 and the ERA-Interim reanalyses (**c**) as a function of time and height.

anomalously warm tropical tropopause temperatures below 20 km (Fig. 7c, d). Westerly shear reduces the tropical upwelling and enhances the horizontal transport and mixing of stratospheric trace gases poleward, consistent with anomalously cold temperature in the extratropical lower stratosphere (Plumb and Bell, 1982; Trepte and Hitchman, 1992; Randel and Wu, 1996; Randel et al., 1999; Randel, 1987; Hamilton, 1998). The magnitude of the QBO westerly wind and shear amplitude variations
5   near the tropical tropopause region in ERA5 corroborates its weaker tropical upwelling in the UTLS compared to ERA-Interim (Figs. 1–3). Conversely, the region of tropical easterly shear between 20 and 30 km is associated with anomalously cold tropical temperatures between 20 and 30 km, leading to enhanced tropical upwelling by the secondary meridional circulation associated with the QBO (Randel et al., 1999; Choi et al., 2002). The tropical upwelling is anti-correlated with the tropical temperature above the tropopause and its strength modulates trace gas mixing ratios by advecting tropospheric air into the stratosphere
10   (Randel et al., 2006; Diallo et al., 2018; Ray et al., 2020). The magnitude of the QBO modulation of the circulation is stronger in ERA5 than in ERA-Interim as shown in both QBO-induced temperature and zonal mean wind amplitude variations for the 1979–2018 period. Near the polar vortex region, the QBO effect on temperatures in ERA5 is larger than in ERA-Interim likely related to the Holton-Tan effect (i.e. the QBO modulates the Arctic polar vortex) (Holton and Tan, 1980; Garfinkel et al., 2012;

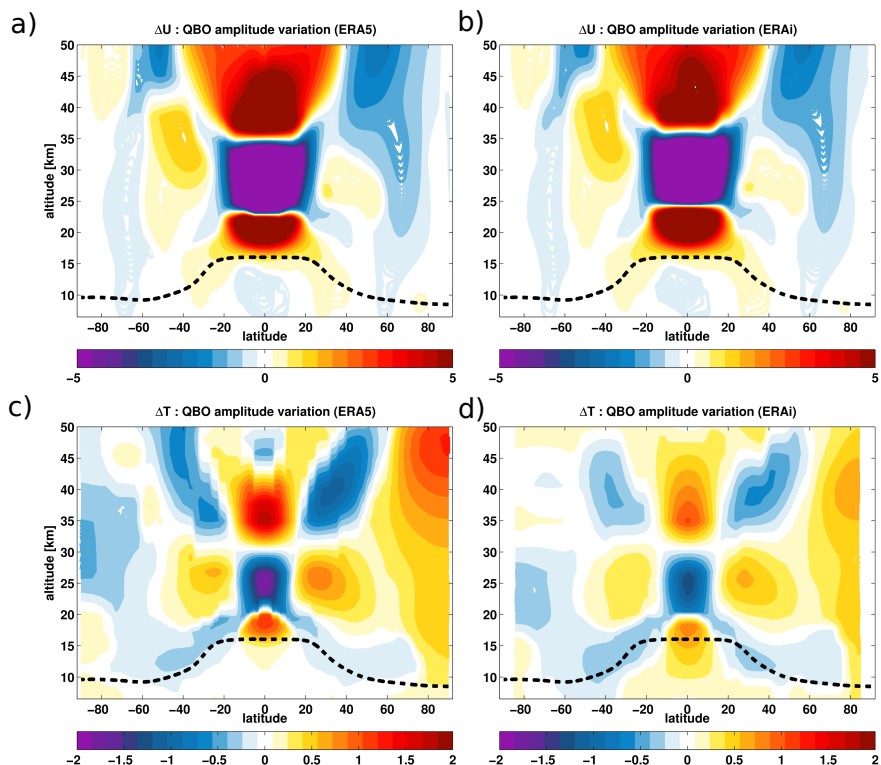

**Figure 7.** QBO amplitude variation derived from the ERA5 and the ERA-Interim zonal mean wind (**a**, **b**) and temperature (**c**, **d**) using the hybrid regression analysis for the 1979-2018 time period. The QBO amplitude variations are estimated by fitting on the QBO predictor using the regression model (10). The units of the QBO amplitude variations are $m \cdot s^{-1}$ and K. In **a** and **b**, the easterly QBO shears and winds are in blue color, while the westerly QBO shears and winds are in red. The climatological tropopause from the ERA-Interim is shown in the black dashed horizontal line.

Lu et al., 2014). In the upper stratosphere above 30 km, large differences in the strength are also visible in both temperatures and zonal mean winds associated with the differences in the QBO westerly shear and wind regions, consistent with the differences in $\overline{w^*}$ (Fig. 1).

In addition to the analysis of the secondary meridional circulation, we analyzed the QBO- and ENSO-induced variability in
5   the stratospheric residual circulation from variations in $\overline{w^*}$ and $\psi^*$. Considered as one of the major modes of BDC modulation (Diallo et al., 2019), the coupled atmosphere-ocean phenomenon, ENSO, is defined as extreme sea surface temperature (SSTs) changes in the tropical Pacific Ocean. These drastic changes in SSTs encompass with severe surface climate and weather conditions (e.g., Bjerknes, 1969; Cagnazzo and Manzini, 2009; Wang et al., 2016). ENSO has two phases known as El Niño (anomalously warm SSTs) and La Niña (anomalously cold SSTs), and its occurrence varies between 2 and 8 years (Philander,
10   1990; Baldwin and O'Sullivan, 1995). El Niño modulates the UTLS by warming the upper troposphere and cooling the tropical

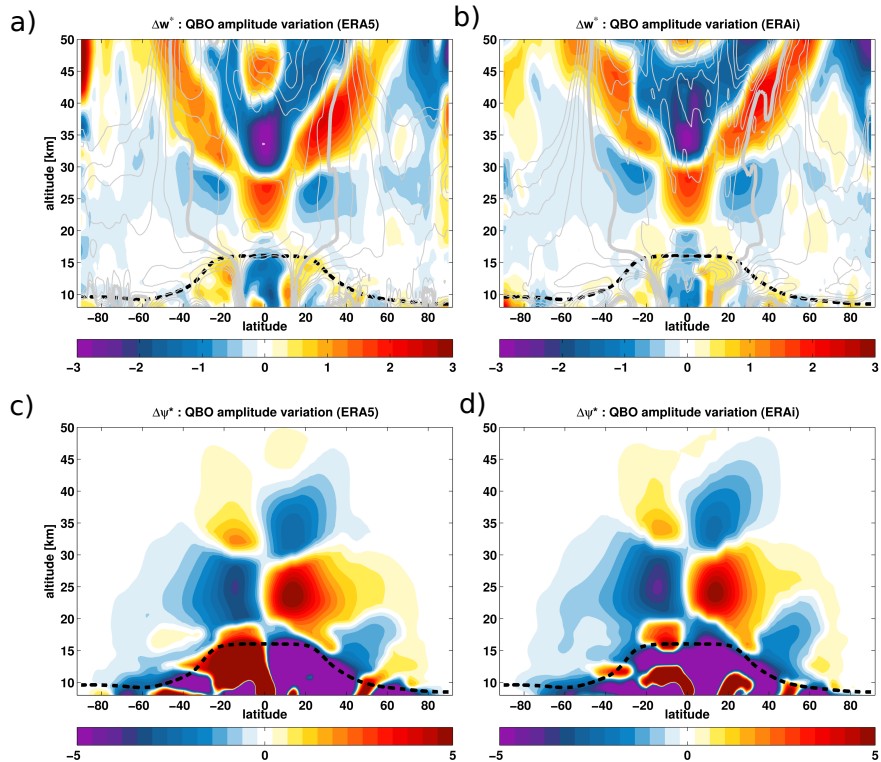

**Figure 8.** QBO impact on the residual vertical velocity ($\overline{w^*}$) (**a**, **b**) and the residual stream-function ($\psi^*$) (**c**, **d**) from the ERA5 (**a**, **c**) and the ERA-Interim (**b**, **d**) reanalyses for the 1979–2018 period. The amplitude of the $\overline{w^*}$ and $\psi^*$ variations attributed to the QBO signal are estimated by fitting on the QBO predictor using the regression model (10) and normalized by the standard deviation (SD) of the proxy for the 1979–2018 period. Unit: $\mathrm{mm} \cdot \mathrm{s}^{-1} \times \mathrm{SD(proxy)}$. The climatological tropopause from the ERA-Interim is shown in the black dashed line. Zonal mean climatology of $\overline{w^*}$ is overplotted as dashed grey lines.

lower stratosphere. The latter has been associated with an acceleration of the ascending branch of the BDC (Randel et al., 2009; Calvo et al., 2010; Konopka et al., 2016).

To quantify the QBO- and ENSO-induced variability in $\overline{w^*}$ and $\psi^*$, the statistical analysis (eq. 10) is performed by explicitly including ENSO and QBO predictors to isolate their modulations of the BDC. The QBO and ENSO coefficients in the regres-
5 sion fit are normalized by the standard deviation of the predictors' proxies for the 1979–2018 period, except for the temperature and zonal mean wind figures, and will be called here the QBO and ENSO *amplitude variations*. This direct approach gives similar results as the differentiating approach of the residuals (Diallo et al., 2017, 2018, 2019).

The results of the QBO-induced variability in $\overline{w^*}$ show a very good agreement between the ERA5 and ERA-Interim reanalyses (Fig. 8a, b). Both reanalyses depict similar patterns of the QBO-induced changes in $\overline{w^*}$. The structural changes in $\overline{w^*}$
10 indicate a clear decrease in the tropical upwelling below 20 km induced by the QBO westerly shear between 15 and 20 km and consistent with QBO-induced anomalously warm tropical temperature changes in tropical UTLS below 20 km (Fig. 7c, d).

Conversely, the tropical upwelling increases between 20 and 30 km due to the QBO easterly shear in that region, consistent with the QBO-induced anomalously cold tropical temperatures between 20 and 30 km (Fig. 3c, d). Another decrease in the tropical upwelling is induced by the QBO westerly shear above 30 km, consistent with the structure of QBO-induced temperature changes (Fig. 7c, d). The $\overline{w^*}$ variations induced by the tropical QBO westerly shear region at altitudes 15-20 km, QBO easterly phase at altitudes 20-30 km and tropical westerly QBO shear above 30 km are consistent with previous findings regarding QBO modulations (Plumb and Bell, 1982; Trepte and Hitchman, 1992; Collimore et al., 2003; Niwano et al., 2003; Punge et al., 2009). Also note that the positive $\overline{w^*}$ anomalies induced by the QBO easterly shear in the tropical middle stratosphere below 30 km, and extending toward the extra-tropics are remarkably well captured in both reanalyses. The QBO westerly shear-induced negative $\overline{w^*}$ anomalies in the middle and upper stratosphere above 30 km agree fairly well in both reanalyses. The main differences between the ERA5 and ERA-Interim reanalyses in term of QBO modulation occurs in the strength of the QBO-induced variability, particularly, with the ERA-Interim showing weaker QBO effects on the BDC. In addition, the positive $\overline{w^*}$ anomalies in the upper tropical stratosphere above 40 km are missing in the ERA-Interim.

Our analysis of the QBO-induced variability in the residual stream-function circulation is also consistent with the QBO impact on $\overline{w^*}$ (Fig. 8a, b). The QBO-induced morphological changes in the residual stream-function show a good agreement between the ERA5 and ERA-Interim reanalyses. The region of QBO westerly shear decreases the residual circulation below the altitude of about 20 km. The regions of QBO easterly shear increase the residual circulation between the altitude of about 20 and 30 km while the QBO westerly shear decreases it above 30 km. The strongest QBO-induced changes in the residual circulation occur between the tropopause and at an altitude of about 40 km with a maximum modulation of the circulation appearing between 20 and 30 km. Also, the residual stream-function shows a stronger QBO modulation in the ERA5 reanalysis than in the ERA-Interim reanalysis (Fig. 8c, d).

The ENSO-induced variations in $\overline{w^*}$ agree well between the two reanalyses in their morphology. The structural changes in $\overline{w^*}$ in the UTLS region are characterized by positive anomalies in the tropics and negative anomalies in the extra-tropics (Fig 9a, b). The El Niño-like condition enhances the tropical upwelling between 15 and 20 km. These ENSO-induced variations in the $\overline{w^*}$ from the two reanalyses agree well. During El Niño-like conditions, the strengthening of the tropical upwelling (positive anomalies of the $\overline{w^*}$ in the tropics) increases upward transport of young air from the troposphere into the stratosphere (Calvo et al., 2010; Konopka et al., 2016; Diallo et al., 2019). The enhanced extratropical downwelling leads to an increase in the downward transport of old stratospheric air into the polar regions, thereby, impacting mid-latitude ozone budgets (e.g., McLandress and Shepherd, 2009; Hegglin and Shepherd, 2009; Lin and Fu, 2013; Butchart, 2014; Hardiman et al., 2014; Neu et al., 2014; Banerjee et al., 2016; Diallo et al., 2019). However, there are differences in the magnitude of the ENSO-induced variation in $\overline{w^*}$. The $\overline{w^*}$ changes related to ENSO show a globally stronger tropical upwelling and extratropical downwelling in the ERA5 reanalysis than in the ERA-Interim reanalysis, except in the tropical UTLS below 20 km where ERA5 exhibits weaker anomalies than ERA-Interim. This specific difference in the UTLS suggests that because of the high vertical resolution in ERA5, the tropospheric ENSO-induced variation is more confined below the tropopause barrier in ERA5 than in ERA-Interim, consistent with larger inter-annual differences in the reanalyses at levels below the cold point (Tegtmeier et al., 2020). The ERA-Interim $\overline{w^*}$ positive anomalies in the inner tropics extend from the upper troposphere into the lower stratosphere

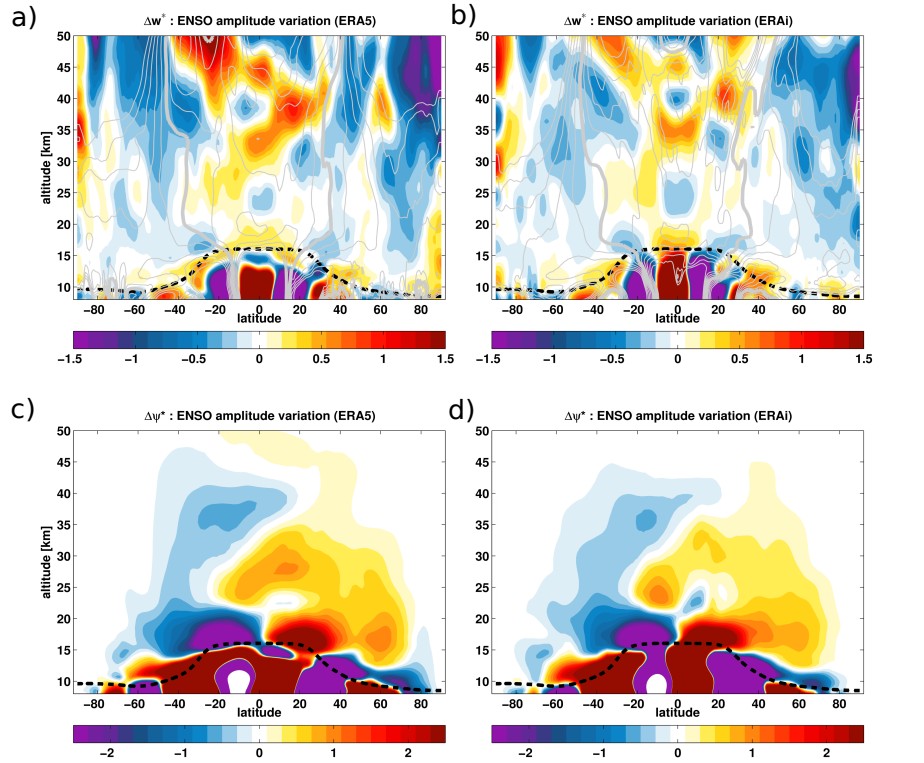

**Figure 9.** ENSO impact on the residual vertical velocity ($\overline{w^*}$) (**a**, **b**) and the residual stream-function ($\psi^*$) (**c**, **d**) from the ERA5 (**a**, **c**) and the ERA-Interim (**b**, **d**) reanalyses for the 1979–2018 period. The amplitude of the $\overline{w^*}$ and $\psi^*$ variations attributed to the ENSO signal are estimated by fitting on the ENSO predictor using the regression model (10) and normalized by the standard deviation of the proxy for the 1979–2018 period. Unit: $mm \cdot s^{-1} \times SD(proxy)$). The climatological tropopause from the ERA-Interim is indicated as the black dashed horizontal line. Zonal mean climatology of $\overline{w^*}$ is overplotted as dashed grey lines.

while in ERA5 this is not visible. Additional differences occur in the tropical middle stratosphere between 20 and 30 km where ERA-Interim shows larger negative anomalies than the ERA5. In the northern hemisphere upper stratosphere, ERA5 also reveals larger negative $\overline{w^*}$ anomalies than the ERA-Interim, which is likely due to the differences in wave activity (Randel et al., 2002, 2008).

5      Insights on the ENSO-induced structural changes in the advective BDC become clearer from the analysis of the residual stream-function (Fig. 9c, d). The structural changes in the residual circulation induced by the ENSO also show a good agreement between the ERA5 and ERA-Interim reanalyses, consistent with the $\overline{w^*}$ changes induced by the ENSO. In both reanalyses, ENSO strengthens the residual circulation of the BDC, particularly the shallow branch (Fig. 9c, d) (Diallo et al., 2019; Yang et al., 2014). The ENSO-induced changes in the deep branch are less evident compared to the shallow and transition branches, 10   nevertheless they are stronger in ERA5 than in ERA-Interim. These spatial changes in the residual circulation is consistent with positive (negative) stream-function changes in the northern hemisphere (southern hemisphere). The apparent differences

between the two reanalyses are associated with the strength of the ENSO-induced acceleration of the BDC. In the inner tropical UTLS below 20 km, ERA5 exhibits weaker circulation anomalies than the ERA-Interim, consistent with the differences between the two reanalyses in the tropical upwelling (Fig. 9a, b). In the middle and upper stratosphere, the strengthening of the residual circulation is stronger in ERA5 than in ERA-Interim. The strengthening of the deep branch in response to ENSO does not extend as far upward in ERA-Interim as it does in ERA5, consistent with less evident ENSO-induced changes in the deep BDC branch (Diallo et al., 2019). As all differences in the circulation and in the natural variability disclose possible discrepancies in the BDC forcings, we analyze the contribution of the planetary and gravity wave in the following (Sect. 3.4).

## 3.4 Planetary and gravity wave forcings

To better understand the contribution of wave forcings to the circulation and the natural variability differences between the ERA5 and ERA-Interim reanalyses, we evaluate the net forcing, the planetary and the gravity wave driving of the BDC (Haynes et al., 1991; Rosenlof and Holton, 1993; Newman and Nash, 2000; Plumb, 2002; Shepherd, 2007). Figures 10a–i show the annual mean climatology of the net forcings of the BDC and the contributions due to the planetary and gravity wave drag together with the associated differences between the ERA5 and ERA-Interim reanalyses for the 1979–2018 time period. Note that the zonal wave forcing $\overline{\Im}$ can induce meridional wind contributions $\delta\overline{v^*}$ (Holton, 1986; Holton et al., 1995). Assuming in the zonal momentum balance (eq. 4) steady state conditions ($\partial\overline{u}/\partial t = 0$) and negligible meridional and vertical gradients of the zonal wind, and assuming that the right hand side of equation (4) is only given by wave forcing $\overline{\Im}$, this equation reduces to $\overline{\Im} = -f * \overline{v^*}$ (Holton, 1986; Holton et al., 1995). This means that the meridional wind contribution $\delta\overline{v^*}$ that is attributed to zonal wave forcing can be written as: $\delta\overline{v^*} = -\overline{\Im}/f$. Correspondingly, positive wave drag indicating eastward forcing weakens the BDC ($\delta\overline{v^*}$ is equatorward), and negative wave drag indicating westward forcing weakens westerly winds, therefore, accelerating the BDC ($\delta\overline{v^*}$ is poleward). (Please note that the Coriolis parameter, $f$, changes its sign at the equator, and this will result in $\delta\overline{v^*}$ switching its sign at the equator if the sign of $\overline{\Im}$ does not switch).

Overall, the climatological structure of the mean wave drag agrees well between the ERA5 and ERA-Interim reanalyses, including the positive wave drag in the easterly zonal mean wind regions and negative wave drag in the westerly zonal mean wind regions (Fig. 10a–f). However, significant differences occur in the amplitude of the wave drag (Fig. 10g–i). The net forcings of the BDC shows that the gravity wave drag contributes the most to the significant lower stratospheric circulation differences between the ERA5 and ERA-Interim. The breaking of synoptic and small-scale waves near the tropical tropopause layer and the equatorward upper flank of the subtropical jets are the primary forcing sources of the transition and shallow branches of the BDC (Plumb, 2002; Shepherd and McLandress, 2011; Diallo et al., 2019). In the UTLS below 20 km, the weaker tropical upwelling in ERA5 compared to ERA-Interim is due to a significantly weaker gravity wave breaking at the equatorial-ward upper flank of the subtropical jets in the ERA5 (Fig. 10g, i). Thus, this difference in the strength of gravity wave breaking in the lower stratosphere explains the weaker ERA5 tropical upwelling as observed in the residual vertical velocity, residual mass stream-function and the RCTTs. The significant contribution of planetary wave is mainly located either below the tropopause or far from the tropical upwelling driving regions, therefore, has a minor contribute to the differences in shallow branch of the BDC between the ERA5 and ERA-Interim reanalyses. In the extratropical upper stratosphere above 30 km, both planetary

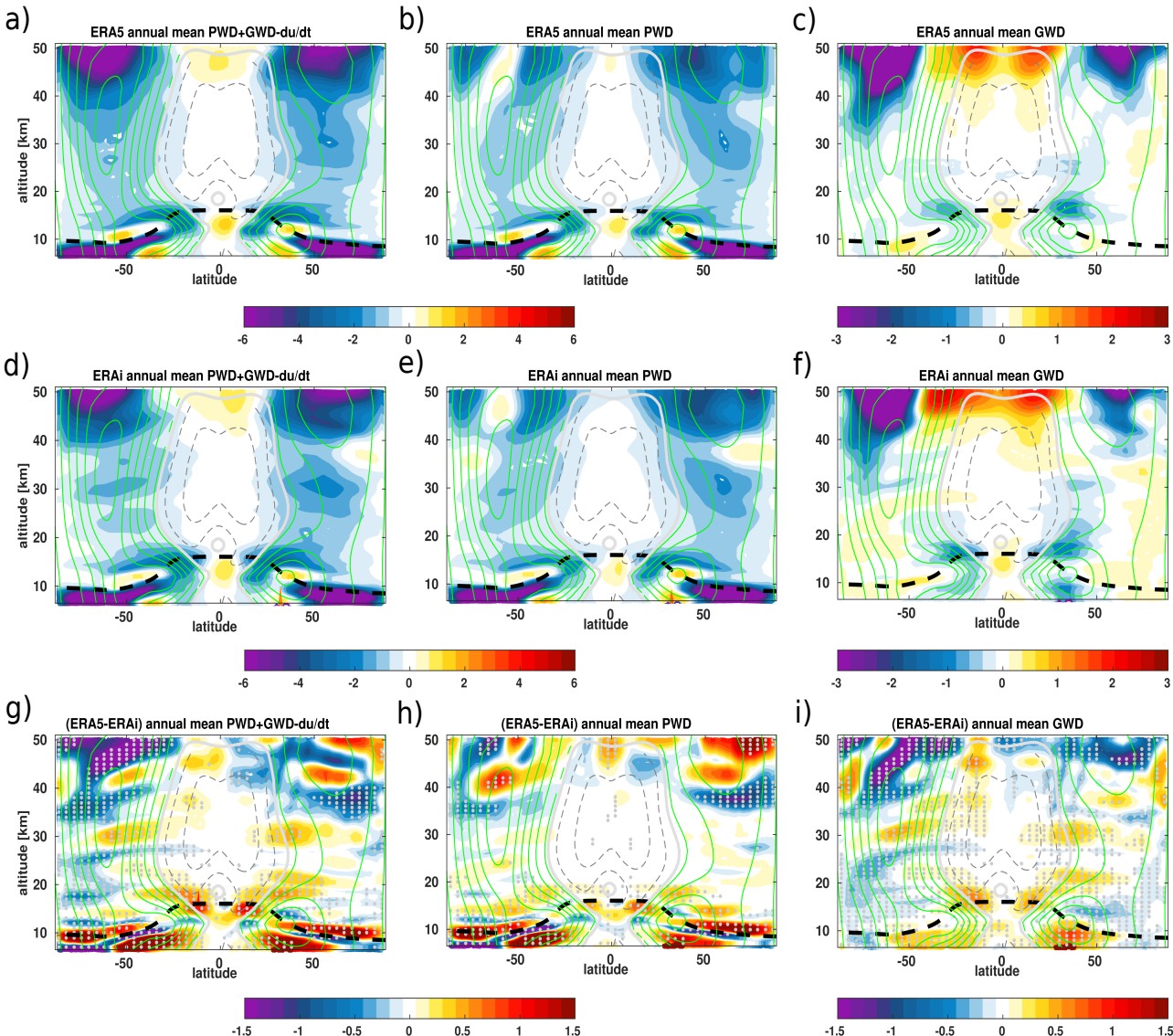

**Figure 10.** Zonal mean distribution of the annual mean net wave forcing (planetary wave drag + gravity wave drag - du/dt) (**a, d**), planetary wave drag (**b, e**) and gravity wave drag (**c, f**) together with the associated differences (**g-i**) between the ERA5 (**a–c**) and ERA-Interim reanalyses (**d–f**) for the 1979–2018 time period. The unit of the wave drag is in m · s$^{-1}$ · day$^{-1}$. The climatological tropopause from the ERA-Interim is indicated as the black dashed horizontal line. The thick white line indicates the zero line zonal mean wind. The thin green and grey lines indicate the climatological zonal mean wind. Grey dots in panels (**g, h, i**) indicate regions where the differences between the ERA5 and ERA-Interim reanalyses are statistically significant at 95% estimated using student t-test.

and gravity wave contribute to the circulation differences between the ERA5 and ERA-Interim reanalyses with a stronger wave forcings in ERA5 than in ERA-Interim (see blue area in Fig. 10g–i). While the planetary wave drag is stronger in the extra-tropics between 30 and 40 km in ERA5 than in ERA-Interim, it is weaker above 40 km. In the tropical upper stratosphere above 40 km, the gravity wave drag in ERA5 is significantly weaker than in ERA-Interim. These differences are likely due to better vertical and horizontal resolution, higher model top and changes to the non-orographic gravity wave drag parametrization scheme in ERA5 (Shepherd et al., 2018; Polichtchouk et al., 2018). In ERA-Interim the non-orographic gravity wave drag was handled by the Rayleigh drag, but in ERA5 the scheme is based on the non-orographic gravity wave parametrization from Warner and McIntyre (2001). The energy deposition from upward propagating gravity waves is a significant contributor to the thermodynamic budget at these altitudes. In addition, note that the sponge layer is also not always applied equally to different variables, and is sometimes enhanced in the vicinity of the equator presumably to control equatorial inertial instability.

Furthermore, the significant differences in wave drag between the ERA5 and ERA-Interim reanalyses reveal seasonal variations (Figs. A1a–i and A2a–i). The contribution of the planetary and gravity wave drag to the lower stratospheric circulation differences is even more clearer in the seasonal means. During boreal winter in the tropical lower stratosphere region, the net wave driving of the BDC in the ERA5 reanalysis shows a weaker gravity wave breaking at the equatorial flank of the subtropical jets than in ERA-Interim (Figs. A1g, i), leading to a significantly weaker ERA5 tropical upwelling. The significant contribution of the planetary wave drag to the circulation differences between ERA and ERA-Interim are stronger below the tropopause and far from the key regions driving the upwelling. Analogously to the annual mean of wave forcings of the BDC (Fig. 10), both planetary and gravity wave breaking contribute to the seasonal circulation differences between the ERA5 and ERA-Interim reanalyses in the upper stratosphere above 30 km. These circulation discrepancies in the upper stratosphere are even stronger during the boreal winter in the both hemispheres, but can be seen in a much larger area in the southern hemisphere. The net wave forcings of the BDC reveal that the planetary and gravity wave breaking are significantly stronger in the southern hemisphere during the boreal winter, which is, in turn, consistent with the reported stronger large-scale downwelling in polar regions (Fig. A1h, i). During boreal summer, the differences in the tropical lower and extratropical upper stratosphere induced by mainly planetary and gravity wave dissipation are also consistent with the discrepancies in the BDC between the ERA5 and ERA-Interim reanalyses. In the lower stratosphere, the gravity wave breaking at the equatorial flank of the subtropical jet causes the significant differences between the ERA5 and ERA-Interim tropical upwelling (Fig. A2a–i). The observed maximum of the seasonal variations in the tropical upwelling at 70 hPa between 0 and 20° N is due to a significantly weaker gravity wave breaking in the ERA5 lower stratosphere. In the upper stratosphere, the gravity wave breaking in the southern hemisphere winter has a stronger contribution than the planetary waves and is also stronger in ERA5 than in ERA-Interim, consistent with the reported hemispheric asymmetry in the $\overline{w^*}$, RCTT, and natural variability induced variations. As the strength of the BDC trends is also driven by the net wave breaking, we expect the BDC trends to be impacted by the weaker wave drag in the ERA5 reanalysis, therefore, we estimate the trends in the following (Sect. 4).

To further link the differences in the upwelling and wave drag, we calculate the DJF and JJA mean stream-function at 70 hPa (∼18.5 km) using the downward control principle (Haynes et al., 1991), described in equation (8). In addition to being commonly used as the reference level for model inter-comparisons of the upwelling strength (Butchart et al., 2010), the 70

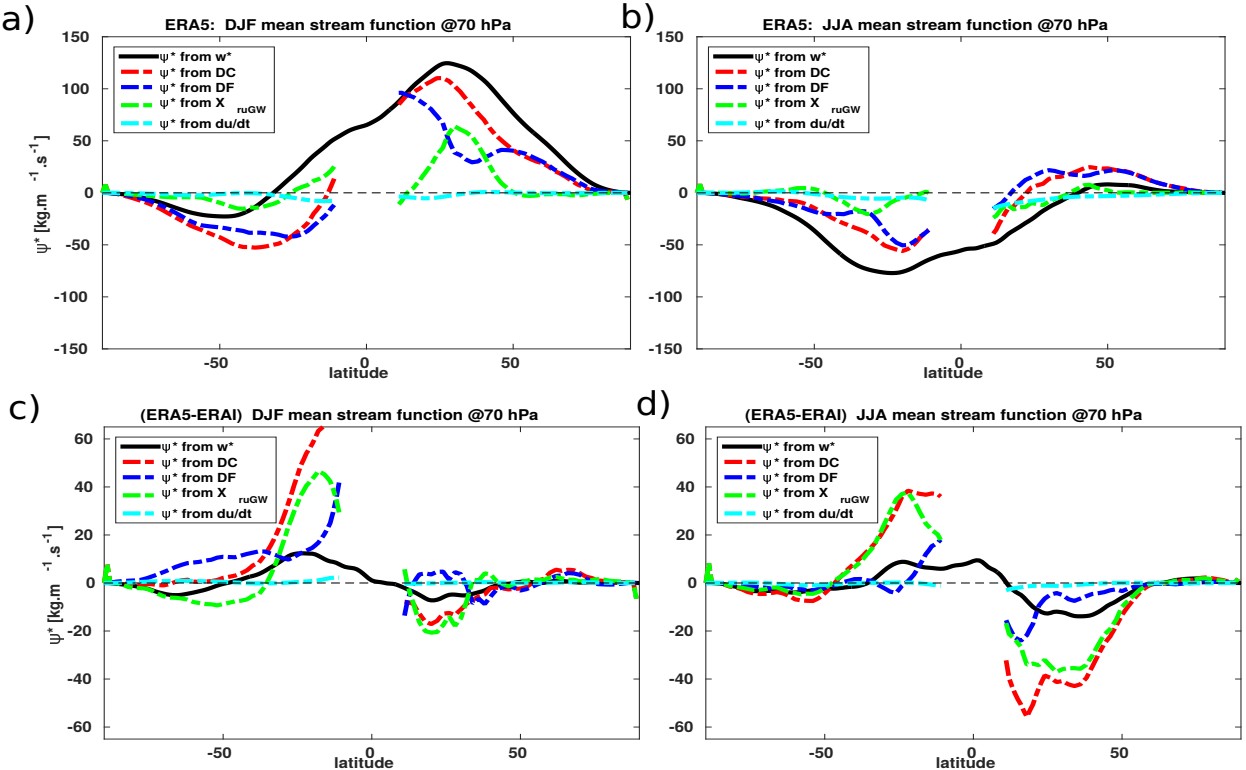

**Figure 11.** DJF (**a**, **c**) and JJA (**b**, **d**) seasonal mean variations of the mass stream-function function ($\psi^*$ in kg $\cdot$ m$^{-1}$ $\cdot$ s$^{-1}$) from the ERA5 (**a**, **b**) and the associated differences between ERA5 and ERA-Interim (**c**, **d**) for the 1979–2018 time period. The $\psi^*$ is calculated from the $\overline{w^*}$, resolved wave drag (DF), unresolved wave drag ($X_{urGW}$) and wind tendency using downward control (DC) principle (Haynes et al., 1991).

hPa pressure level works best for RCTT and age of air in inter-model correlations for almost all the stratosphere (Dietmüller et al., 2017). Figure 11 shows the DJF and JJA mass stream-function seasonal variation at 70 hPa calculated from the ERA5 and ERA-Interim $\overline{w^*}$, resolved wave drag (DF) and unresolved wave drag ($X_{urGW}$) for the ERA5 reanalysis together with the differences between ERA5 and ERA-Interim for the 1979-2018 time period. Clearly, the gravity wave drag is the main driver
5  causing the weaker tropical upwelling in ERA5 compared to ERA-Interim with largest effects occurring within the tropics (Fig. 11 c, d). Due to small differences between ERA5 and ERA-Interim in the contribution of the resolved wave drag (DF), including the resolved planetary and gravity wave drag, we can also conclude further that the upwelling differences between the two reanalyses in term of the gravity wave drag results from the unresolved gravity wave contribution, i.e. the parameterized and imbalance gravity wave drag (Alexander and Rosenlof, 1996; McLandress et al., 2012; Ern et al., 2014). This suggests
10  that the improvement in parameterized non-orographic gravity wave drag, which impacts tropical regions where sub-grid-scale gravity waves from convective sources are dominant, may certainly be the reason of these differences (Chun et al., 2004), consistent with the upwelling and wave drag differences. According to Butchart et al. (2010), the state-of-the-art chemistry-

climate models show a good representation of the resolved wave contribution to driving the annual mean tropical upwelling at 70 hPa with a contribution of 70.7%from resolved waves, 21.1% from orographic gravity wave drag (parametrized in ERA5 and ERA-Interim) and 7.1% from non-orographic gravity waved drag (not included in ERA-Interim but include in ERA5). With a contribution of about 30% according to climate model mean, the differences in the unresolved gravity wave drag between the two reanalyses is consistent with the contribution range to the upwelling. Nevertheless, one should keep in mind that the representation of gravity waves in reanalyses and climate models remains not well known (Seviour et al., 2011; Shepherd, 2014). The uncertain aspect of the reanalyses related to the representation of unresolved (sub-grid scale) processes such as gravity-wave drag, convection, and boundary-layer physics have been improved compared to the ERA-Interim (Hersbach et al., 2020).

## 4 Advective BDC trends

For better insights in the impact of stronger gravity and planetary wave drag on the BDC in ERA5, we also investigate the structural changes in the advective BDC and wave drag for the 1979–2018 time period (Fig. 12a–d). The ERA5 reanalysis shows a long-term acceleration of the BDC, consistent with all reanalysis data sets, except the CFSR reanalysis (Diallo et al., 2012; Abalos et al., 2015; Miyazaki et al., 2016; Chabrillat et al., 2018; Ploeger et al., 2019), and with long-term climate model simulations (e.g., Butchart et al., 2010; Hardiman et al., 2014). The annual mean trend of the residual circulation mass stream-function shows a positive trend in the northern hemisphere and weakly negative trend in parts of the southern hemisphere, indicating a not significant strengthening of the shallow and deep branches of the BDC in both hemispheres (Fig. 12a). The negative BDC trend in the inner tropical UTLS suggests a weakening of the transition branch probably linked to the tropopause rise induced by the combined effect of tropospheric warming by greenhouse gases and stratospheric cooling by ozone depletion (Randel et al., 2000; Santer et al., 2003; Seidel and Randel, 2006; Son et al., 2009; Vallis et al., 2015; Oberländer-Hayn et al., 2016; Šácha et al., 2019; Eichinger and Sacha, 2020). The decomposition of the wave drag trend into planetary and gravity wave forcing provides an estimate of the impact of these two wave forcings on the structural changes in the transition, shallow and deep branches of the BDC. The BDC trend in ERA5 is induced by the combined contribution of the gravity and planetary wave breaking (Fig. 12c, d). The negative trends in the net wave forcings of the BDC between 35 and 45 km indicate an intensified planetary and gravity wave breaking in both hemispheres (Fig. 12b–c). While the shallow branch is accelerated mainly by enhanced gravity wave breaking in the tropical and subtropical lower stratosphere, the deep branch is driven by a sum of the contribution from the planetary and gravity wave breaking at high altitudes. The planetary wave breaking seems to contribute the most to the acceleration of the deep branch of the BDC, consistent with previous studies (Sigmond and Shepherd, 2014; Abalos et al., 2015; Šácha et al., 2019). The hemispheric asymmetry in the BDC trends is due to the asymmetry in wave breaking, which is stronger in the northern hemisphere than in the southern hemisphere as indicated by the negative net forcing trend (Fig. 12b). Also note that the southern hemispheric circulation cell exhibits an apparent structural difference compared to the northern hemispheric circulation cell. A detailed analysis of the BDC trends is presented in further studies using the

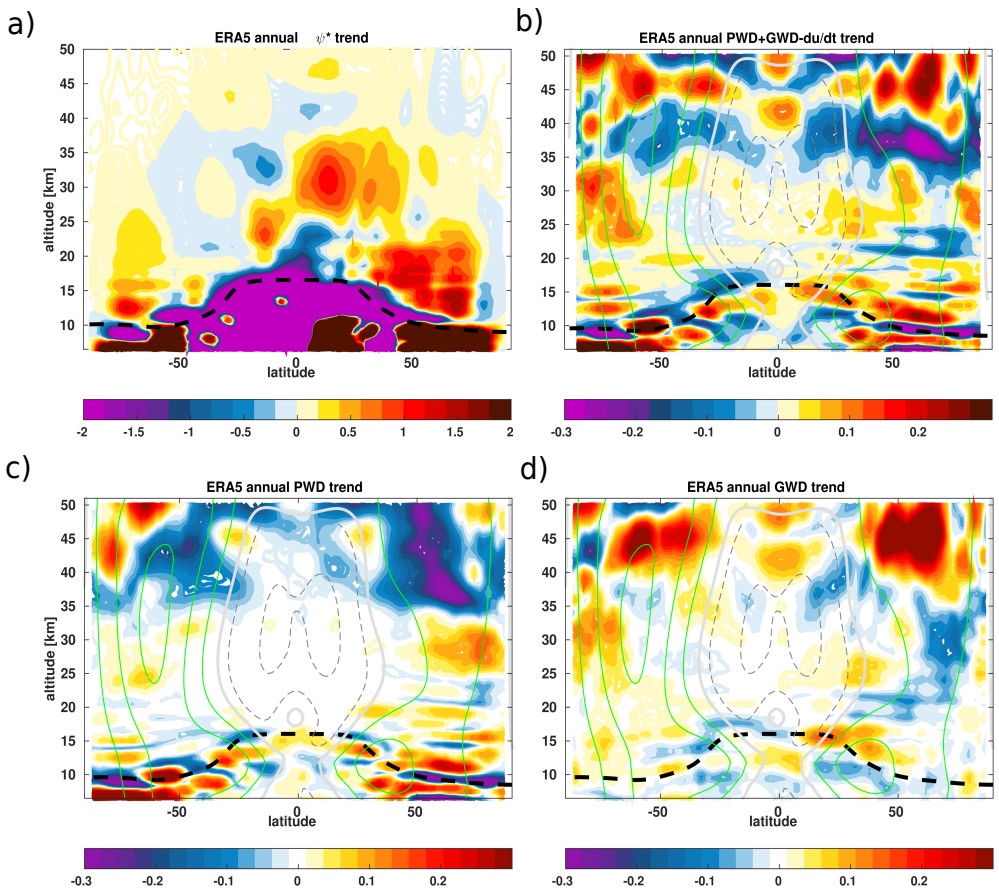

**Figure 12.** Linear trends of the $\psi^*$ (**a**), net wave forcing (**b**), planetary wave drag (**c**) and gravity wave drag (**d**) from the ERA5 reanalysis for the 1979–2018 time period. The units of the $\psi^*$ trend and wave drag trend are in $\text{m}^{-2} \cdot \text{s}^{-1} \cdot \text{decade}^{-1}$ and $\text{m} \cdot \text{s}^{-1} \cdot \text{day}^{-1} \cdot \text{decade}^{-1}$, respectively. Trends are statistically significant at 95% estimated using student t-test. The climatological tropopause from the ERA-Interim is indicated as the black dashed horizontal line. The thick white line indicates the zero line zonal mean wind. The thin green and grey lines indicate the climatological zonal mean wind.

stratospheric age of air and its spectrum (Ploeger et al., 2021). Note that the ERA-Interim still shows a negative trend in the residual stream-function (not shown), consistent with the weakening upwelling trend estimated from $\overline{w^*}$ (Seviour et al., 2011).

In addition, the time series of deseasonalized monthly mean tropical upwelling mass flux averaged between the turnaround latitudes shows a comparable inter-annual variability between the ERA5 and ERA-Interim reanalyses at the pressure level of 70 hPa (Fig. 13). The estimated annual mean upwelling mass flux is about $6.17 \times 10^9 \, \text{kg} \cdot \text{s}^{-1}$ for ERA5 with a standard deviation of 0.71 and about $6.86 \times 10^9 \, \text{kg} \cdot \text{s}^{-1}$ for ERA-Interim with a standard deviation of 0.99 for the 1979–2018 period, consistent with the multi-model mean value of $5.9 \times 10^9 \, \text{kg} \cdot \text{s}^{-1}$ from previous studies (e.g, Butchart et al., 2010; Hardiman

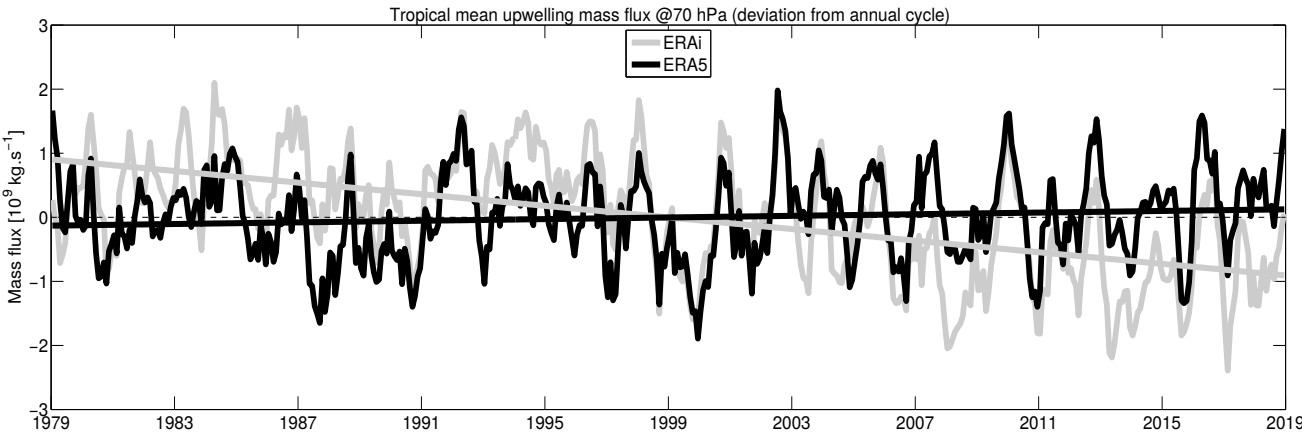

**Figure 13.** Time series of deseasonalized annual mean tropical upwelling mass flux at 70 hPa (∼19 km ) from the ERA5 and the ERA-Interim reanalyses for the 1979–2018 time period. Shown are ascending flux from the ERA-Interim reanalysis (Light grey), ascending flux from the ERA5 reanalysis (Black), linear trend of the ascending flux from the ERA-Interim (Light grey) and linear trend of the ascending flux from the ERA5 reanalysis (Black) for the 1979–2018 time period.

et al., 2014; Linz et al., 2017). The ERA5 tropical upwelling mass flux trend at 70 hPa calculated from $\overline{w^*}$ for the 1979-2018 time period is about 1.5 % per decade, consistent with the observed BDC changes of about 1.7 % per decade (Linz et al., 2017; Fu et al., 2019) and climate model predictions of an increase in the strength of the BDC of about 2 % per decade due to the increasing GHGs in the atmosphere (Butchart et al., 2010; Garny et al., 2011; Hardiman et al., 2014; Eichinger and Sacha, 2020). While the climate model projected trend is statistically significant, the mass flux trend in ERA5 is not statistically

significant at 95% using student t-test with two tail distribution. Note that the reported negative long-term trend by Seviour et al. (2011) in the tropical upwelling mass flux estimated from the ERA-Interim $\overline{w^*}$ is not consistent with the ERA5 trend. Therefore, the ERA5 $\overline{w^*}$ trend is similar to the trend from climate models, which implies that the ERA5 $\overline{w^*}$ estimated with the standard TEM formula can be used as a proxy for climate model validation, which was not the case for the ERA-Interim because of a questionable $\overline{w^*}$ (Seviour et al., 2011). This negative long-term trend is only present in the ERA-Interim reanalysis

when using the standard TEM formula to calculate the $\overline{w^*}$ and is absent in other reanalyses (Abalos et al., 2015; Miyazaki et al., 2016).

As a basis for verification, we also calculated the long-term trend in $\overline{w^*}$ in ERA5. This estimated mean upwelling mass flux trend from the ERA5 reanalysis is about $0.1\pm 0.053 \times 10^9 \, \text{kg} \cdot \text{s}^{-1} \cdot \text{dec}^{-1}$ instead of $-0.47\pm 0.53 \times 10^9 \, \text{kg} \cdot \text{s}^{-1} \cdot \text{dec}^{-1}$ for the ERA-Interim. The relative difference between the two reanalyses in the mass flux is not large, but significant and is between

10 % and 20 % below 70 hPa, indicating that the ERA5 upwelling change is more consistent with observed and climate model predicted strengthening BDC than ERA-Interim.

## 5    Summary and Conclusions

In this study, we assess the climatology, inter-annual variability and trends of the advective stratospheric BDC in the ERA5 and ERA-Interim reanalyses using different circulation metrics, including the $\overline{w^*}$, residual stream-function and RCTT. We also evaluate the impact of the natural variability, in particular the QBO and the ENSO, on the advective BDC as well as the impact

of planetary and gravity wave drag on the advective BDC and its trend.

In our comparisons of the circulation from the ERA5 reanalysis with the ERA-Interim reanalysis, we found a good agreement in the morphology of the advective BDC, including the tropical upwelling and extratropical downwelling as well as in the QBO- and ENSO–induced impact. Despite the good agreement in the spatial structure, there are significant differences between the ERA5 and the ERA-Interim reanalyses in the strength of the advective BDC and in its modulation by the natural variability

in the UTLS and upper stratosphere regions. The slower advective BDC and its strong modulations by the natural variability in the ERA5 reanalysis is due to weaker planetary and gravity wave drags. In the tropical pipe region below 20 km, the tropical upwelling is up to 40 % weaker in ERA5 than in ERA-Interim mainly due to a significantly weaker gravity wave breaking at the equatorial flank of the subtropical jet. The differences in planetary wave drag between the two reanalyses are significant, but weaker than the differences in the gravity wave drag at key regions of upwelling forcings, therefore, they have

minor contribution to the slower upwelling differences. In the extra-tropics, the large-scale downwelling near the polar vortex is stronger in ERA5 than in ERA-Interim linked to significant differences in planetary and gravity wave drag, which might impact on the representation and strength of the polar vortex. These differences in ERA5 vary seasonally and are consistent within all metrics used for evaluating the BDC, including the $\overline{w^*}$, $\psi^*$, RCTT and modulations by the natural variability. In addition, the differences between the ERA5 and ERA-Interim reanalyses show a hemispheric asymmetry and a seasonal variation with

a maximum difference of about 60 % in downelling (stronger in ERA5) in the southern hemisphere.

Regarding the QBO signal in the zonal mean wind and temperatures, we found a good agreement in the morphology of the secondary meridional circulation between the two reanalyses. The analysis of the QBO amplitude variation revealed anomalously warm tropical tropopause temperatures associated with the QBO westerly shear modulation, which decreases the tropical upwelling in both reanalyses. Conversely, the QBO easterly shear modulation induced anomalously cold tropical temperatures,

leading to enhanced tropical upwelling by the QBO secondary circulation. Similarly to the QBO-induced secondary meridional circulation in the temperatures, the regression analysis of the QBO-induced structural changes in $\overline{w^*}$ and $\psi^*$ shows a very good agreement in the modulations of the BDC between the two reanalyses. Both reanalyses agree remarkably well on a weaker upwelling in the UTLS region below 20 km and in the upper stratosphere above 30 km due to the QBO easterly and westerly shear modulations. Besides the good structural agreement, noticeable differences are also found in the strength of the QBO

easterly and westerly shear modulations. Between 15 and 20 km, the QBO phases in ERA-Interim are stronger than in ERA5, while above 20 km, the QBO modulation is stronger in ERA5 than in ERA-Interim.

The regression analysis of the ENSO-induced structural changes in $\overline{w^*}$ and $\psi^*$ shows a very good agreement in the modulations of the BDC between the two reanalyses. However, significant differences have also been found in the strength of the advective BDC modulations by ENSO. Analogously to the QBO effects, the ENSO impact on the BDC is globally stronger in

ERA5 than in ERA-Interim, except in the UTLS below 20 km where ERA5 seems to be weaker than ERA-Interim, consistent with the weaker ERA5 tropical upwelling.

The analysis of the planetary and gravity wave drag clearly shows that the main differences in the advective BDC in the UTLS region between the ERA5 and ERA-Interim reanalyses, including the tropical upwelling and mid-latitude downwelling, are mainly due to a weaker gravity wave breaking at the equatorial flank of the subtropical jets in ERA5. The differences in the planetary wave drag between the ERA5 and ERA-Interim reanalyses are significant, but weaker than the differences in gravity wave drag in the lower stratosphere, therefore, not dominant. In the upper stratosphere above 30 km, the differences in gravity and planetary wave drag between the two reanalyses are significantly large, consistent with a weaker BDC in the UTLS and stronger BDC in the middle and upper stratosphere in ERA5. These differences in wave forcings, in particular, in the BDC strength and its modulations, are very likely related to a combination of several factors, including a higher resolution, higher model top, new non-orographic gravity wave parameterization scheme. In addition, progress has also been made in the more realistic representation of the sponge layer in ERA5, which is presumably designed to damp upward propagating waves near the model top to avoid reflections from the upper boundary (Hersbach et al., 2020).

Finally, the estimates of the advective BDC trend in ERA5 show a global acceleration of the annual mean residual circulation stream-function of about 1.5 % per decade for the 1979-2018 period. Althrough not statistically significant at 95%, this acceleration rate is consistent with observed acceleration of the BDC of about 1.7 % per decade (Thompson and Solomon, 2005; Linz et al., 2017; Fu et al., 2019) and with future climate model predicted rate of the strength of the BDC of about 2 % per decade due to the increasing level of atmospheric GHGs (Butchart et al., 2010; Garny et al., 2011; Hardiman et al., 2014). This implies a recommended use of ERA5 $\overline{w^*}$ as proxy for climate model validations instead of the ERA-Interim. The strengthening of the BDC is induced by the increasing planetary and gravity wave breaking in both hemispheres above 30 km. The trends in the residual circulation stream-function are stronger in the northern hemisphere than in the southern hemisphere, consistent with the trends of the net wave forcing of the BDC. This hemispheric asymmetry in the wave breaking trends is likely linked to more disturbed winter polar vortex in the northern hemisphere than in the southern hemisphere as well as to ozone recovery in the southern polar vortex (Polvani et al., 2018).

Our findings suggest that the advective BDC from the kinematic ERA5 reanalysis is well suited for climate model validation in the UTLS and mid-stratosphere when using the standard formula of zonally-averaged zonal momentum equation. The reported differences between the two reanalyses may also affect the nudged climate model simulations. According to recent findings, nudged climate models show slightly stronger upwelling compared to the free-running simulations and exhibit marked differences compared to the direct estimates of the upwelling from the reanalysis (Chrysanthou et al., 2019; Davis et al., 2020). Therefore, additional studies are needed to investigate whether or not nudging climate models toward ERA5 reanalysis will accurately reproduce the tropical upwelling trends comparable to the free-running versions and to directly estimated residual circulation from the ERA5 reanalysis. Finally, studies seeking for a better understanding of the impact of the new non-orographic gravity wave parameterization scheme, higher model top, and the representation of the sponge layer in ERA5 on the differences in the upper stratosphere and polar regions are also needed.

**Appendix A: Seasonal variations of planetary and gravity wave forcings**

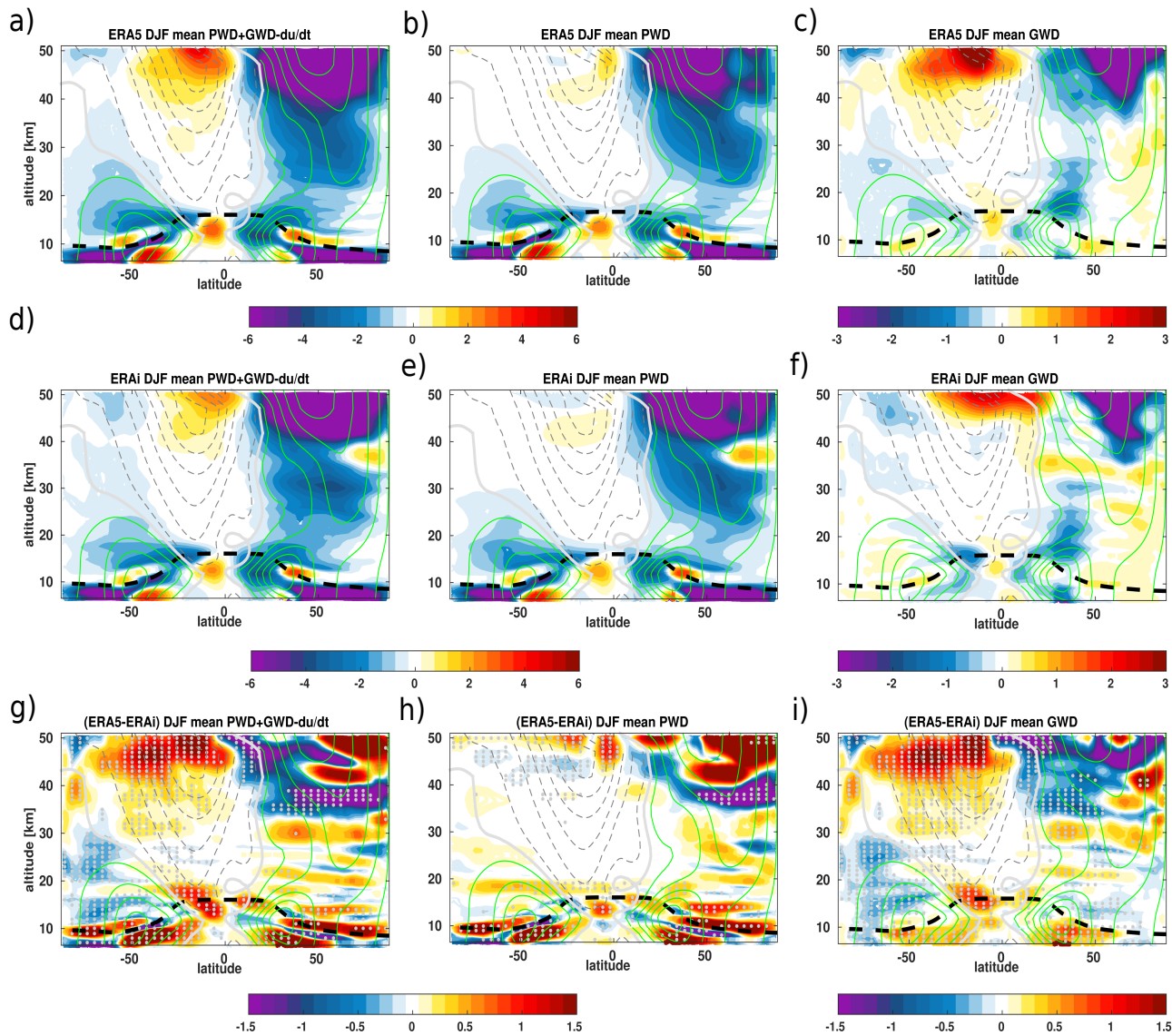

**Figure A1.** Zonal mean distribution of the winter seasonal mean (DJF) net wave forcing (planetary wave drag + gravity wave drag - du/dt) (**a**, **d**), planetary wave drag (**b**, **e**) and gravity wave drag (**c**, **f**) together with the associated differences (**g-i**) between the ERA5 (**a–c**) and ERA-Interim (**d–f**) reanalyses for the 1979–2018 time period. The unit of the wave drag is in m · s$^{-1}$ · day$^{-1}$. Grey dots in panels (**g**, **h**, **i**) indicate regions where the differences between the ERA5 and ERA-Interim reanalyses are statistically significant at 95% estimated using student t-test. The climatological tropopause from the ERA-Interim is indicated as the black dashed horizontal line. The thick white line indicates the zero line zonal mean wind. The thin green and grey lines indicate the climatological zonal mean wind.

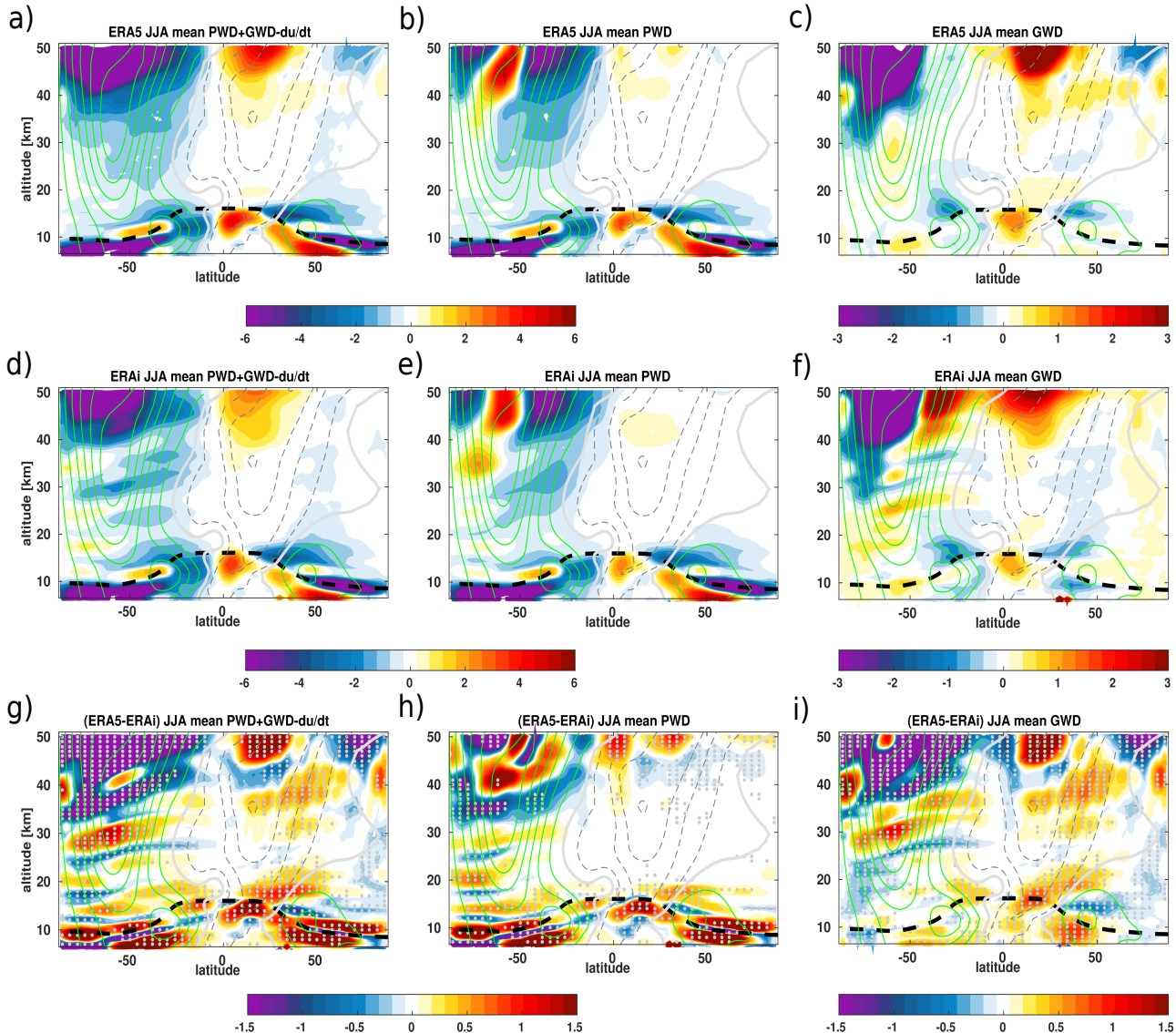

**Figure A2.** Zonal mean distribution of the summer seasonal mean (JJA) net wave forcing (**a, d**), planetary wave drag (**b, e**) and gravity wave drag (**c, f**), together with the associated differences (**g-i**) between the ERA5 (**a-c**) and ERA-Interim (**d–f**) reanalyses for the 1979–2018 time period. The The unit of the wave drag is in $m \cdot s^{-1} \cdot day^{-1}$. Grey dots in panels (**g, h, i**) indicate regions where the differences between the ERA5 and ERA-Interim reanalyses are statistically significant at 95% estimated using student t-test. The climatological tropopause from the ERA-Interim is indicated as the black dashed horizontal line. The thick white line indicates the zero line zonal mean wind. The thin green and grey lines indicate the climatological zonal mean wind.

*Data availability.* The ERA5 reanalysis (https://apps.ecmwf.int/data-catalogues/era5/?class=ea last access: 20 August 2020) and ERA-Interim data (https://apps.ecmwf.int/data-catalogues/era-interim/?class=ei, last access: 20 August 2020) are available. The advective BDC and ERA5 data used for this paper may be requested from the corresponding author (m.diallo@fz-juelich.de).

*Author contributions.* MD designed the study, conducted research, performed the calculation and the complete analysis of the advective
BDC diagnostics as well as drafted the first manuscript. FP calculated the RCTTs. ME calculated the wave decomposition. FP and ME provided helpful discussions and comments. MD edited the final draft with contributions from all co-authors for communication with the journal.

*Competing interests.* The authors declare that they have no conflict of interest.

*Acknowledgements.* MD research position is funded by the Deutsche Forschungsgemeinschaft individual research grant number DI2618/1-1
and Institute of Energy and Climate Research, Stratosphere (IEK-7), Forschungszentrum in Jülich during which this work had been carried out. FP is funded by the Helmholtz Association under grant number VH-NG-1128 (Helmholtz Young Investigators Group A-SPECi). The work by ME was supported by the German Federal Ministry for Education and Research (Bundesministerium für Bildung und Forschung, BMBF) project QUBICC, grant number 01LG1905C, as part of the Role of the Middle Atmosphere in Climate II (ROMIC-II) program of BMBF. We thank Bernard Legras for providing the advective BDC script. We also thank Inna Polichtchouk for providing useful comments
on the draft. We also acknowledge support by the ciclad cluster in Paris, where this work had been carried out. We gratefully acknowledge the Earth System Modelling Project (ESM) for funding this work by providing computing time on the ESM partition of the supercomputer JUWELS at the Jülich Supercomputing Centre (JSC). Moreover, we particularly thank the European Centre for Medium-Range Weather Forecasts for providing the ERA5 and ERA-Interim reanalysis data.

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
