# Peer review of "The advective Brewer-Dobson circulation in the ERA5 reanalysis: climatology, variability, and trends"

_Atmospheric Chemistry and Physics, 2020_

## Referee Comment (RC1) · Roland Eichinger (Referee) · 22 Oct 2020

**Review on:**
**"The advective Brewer-Dobson circulation in the ERA5 reanalysis: re-analysis and trends", ACPD, 2020, by Diallo, M. et al..**

In their paper, Diallo et al. compare several variables which are used to analyse the BDC of the new ERA5 reanalysis data with ERAinterim data. The authors show that the structures of the ERA5 BDC pretty much resemble those of ERAi, but also that there are nuanced differences that are important to consider. Some of the points, like the fairly large difference of tropical upwelling in the lower stratosphere and the clear

difference in the trends can become crucial for future interpretations of stratosphere dynamics and transport. The paper is well-written, it is relevant and important for research on stratospheric dynamics and it fits for ACP.

I am very much in favour of publication of the article in ACP, however, only after several major and a number of minor revisions that I list below. In my opinion, in the paper the authors oftentimes conclude too quickly, without having considered all possibilities and processes. This concerns also one of the major conclusions, where the authors state that tropical upwelling is 40% smaller in ERA5 due to gravity wave forcing (see below). Moreover, the length of the paper could easily be reduced (I give advice below), and at one point the paper loses its main focus (the comparison of ERA5 and ERi) without a good reason. I also strongly suggest to include statistical significance tests for proper interpretation of the results, to somewhat revise the conclusion section and to slightly change the title. Please see my major and minor points listed below and also consider the technical issues that I list at the bottom.

Best wishes
Roland

**Major issues:**

- P1L12-13 and P25L12-13: I did not believe this statement when I first read the abstract and I still don't do so having read the whole paper. Commonly the contribution of gravity waves on tropical upwelling is around 30% here (see Butchart et al. 2010). Stating that a weaker GW forcing reduces tropical upwelling by 40% does not go together with that. The statement seems to base on the sentence "The contribution of the planetary waves to the tropical upwelling differences is less evident" on P21L1-2 (and P26L23), which you use to entirely disregard any PW contribution or anything else. I think what could help to separate the contributions of planetary and GW waves here could be a downward control (Haynes et al. 1991) analysis, but on the basis of patchy Fig. 10, the statement seems adventurous to me. Moreover, how well do the tropopauses fit together between ERAi and ERA5? The differences seem strongly altitude-dependent. This and also Fig. 2b made me think of a possible vertical shift between the reanalyses, that could contribute to the upwelling differences, too.

- In all plots where you calculate the difference between ERAinterim and ERA5 you need to add the statistical significance of the differences. In some regions and/or for some variables, the variability may be so large that the mean differences are not meaningful or small differences might be overseen despite their significance. Please add significance to all those plots and revise your text accordingly. In many occasions (e.g. P26 L16,L24,31) you even mention significance, but it has never been shown.

- P7L26-27 and P25L13-14: "suggesting a stronger advective BDC" That statement seems much too general and in some sense even wrong, because you already showed, that in the tropics, the upwelling is weaker. Often tropical upwelling is used as a proxy for the advective BDC in the whole stratosphere.
As the stronger downwelling is limited to the high latitudes, this might be related to the polar vortex strength, which has some impact on high latitude downwelling. But the polar vortex differences between ERAi and ERA5 are not discussed in the paper. I am not saying you should analyse it, and include plots, but there should be a discussion about it, separating the different regions. The topic only appears very briefly in the very last sentence of the paper. In my opinion this is much too late and too briefly discussed, it should be included properly in the analysis and discussion.

- The paper is quite long and not every analysis is really needed for the conclusions. I think some of the figures (or panels) do not contribute anything to the

final conclusions. I therefore suggest to go through the paper starting from the back, analysing each conclusion with respect to which figures are really needed for that. The rest can be banished to a supplement, where I would also move the figures that now are in the appendix.

For example: P25L14-18: If this is all the outcome from the seasonal climatologies, I think the analysis of the seasonal climatologies can be reduced drastically, and most of the figures can be moved to a supplement. Also, I am not sure if Fig. 3 is needed at all for the conclusions.

- Until section 4, the paper always focuses on the comparison between ERAi and ERA5. In section 4, at first only the ERA5 trends (streamfct and wave drag trends) are analysed, and then the mass flux trend is compared between ERA5 and ERAi again. I do not understand why you do not stick to the comparison between the reanalysis products, as it seemed to me that this is the focus of the paper. I suggest to include the ERA5-ERAi streamfct and wave drag comparison here as well, to keep the theme of the paper. Moreover, I suggest to reflect that theme also in the title of the paper, maybe something like:

  ”Comparison of the advective Brewer-Dobson circulation between ERA5 and ERAinterim reanalysis: climatology, variability and trends“.

  And I particularly suggest to add ”climatology“, because that is what most of the paper deals with (annual and seasonal climatologies).

- The discussion and conclusion section almost completely lacks the connection to literature and it also does not point out what the implications of the study are. I think both these points can easily be addressed. The literature is already outlined nicely in the introduction. The implications can include what the present study means for older conclusions about the BDC based on ERAi data (or model simulation results that are nudged or prescribed to ERAi dynamics), and what now can be assessed better, or more precisely, or differently with ERA5. This would be nice for closing remarks of the paper.

**Minor issues:**

- P3L11: "the strength of the BDC". Do you mean the change of the strength here or really the strength? If the latter, I think the statement is out of place here, since you talk about trends in this paragraph.

- P3L13: remove "reanalyses and for". There were no reanalyses analysed in that study.

- P4L11-12: I assume that the ability to better resolve (macro-scale) meteorological features is mainly du to the higher resolution, and not "apart from that". Hence, please reformulate the sentence.

- Sect. 2.2: Please consider the QJRMS paper by Eichinger and Sacha, (2020). You have to be very cautious with which $w^*$ you use for your analyses. As you do all your calculations in log-pressure coordinates, you may make a mistake when you calculate the $w^*$ trends because in log-pressure coordinates, the trends of the pressure levels (i.e. the temperature trends and thereby the trend in the scale-height, which you implicitly neglect using the log-pressure formulae) is not considered. In that paper, we propose using the goepotential coordinates. However, for your streamfunction trends, you should be fine, because the H cancels out when you use the log-pressure density (Sect. 4.2 of the paper). But you also calculate some $w^*$ trends at the end where it certainly matters. And I am not exactly sure if it is important for the variability calculations. Please make sure your analyses do not include the error.

- P7L4: Remove the overbar over $w^*$, it is not zonal mean here. But throughout the paper, use the overbar when zonal mean is shown and no overbar if it is general. And note somewhere that the overbar stands for zonal mean.

- Fig. 2: Why 15·$\sigma$? The 15 seems very arbitrary, can you explain why you chose it? And why not adding such a measure of uncertainty also to a,b,c? It would be helpful when interpreting the results (see also my major point about statistical significance)

- P10L9: But according to Fig. 2e, that is only true between 35 and 40 km, not everywhere above 20 km.

- P11L14-16: Note that ... polar regions. I do not understand what you try to say with that sentence. I suggest to remove it.

- P11L19-24: The description of the differences is unclear to me:
  - In DJF, I only see a small negative patch in the UTLS and more negative fields in the SH mid to high latitudes. The conclusion in L21 therefore bases on wrong facts.
  - I also do not see the positive differences between 40S and 0, where are you there? Still in DJF?
  The differences generally barely exceed the tropopause, or are just much smaller in the stratosphere, as the mass streamfunction is much smaller there too. So the color bar is maybe not really suitable, and moreover, it would be very helpful here to see statistical significance.
  The statement that the residual circulation is stronger in ERA5 seems wrong. Firstly, you already showed that upwelling is weaker, secondly Fig. 4c shows the opposite.
  Please clarify and correct these points.

- P11L25-27: I do not understand what you mean by "horizontal shift here. Can you elaborate on this please.

- P14L23: "stronger westerly and easterly shear" I can not quite see that. Can you please describe where/ guide the eye a bit?

- P14L25: "into the troposphere below the tropopause .... does not propagate that far downward"
  I can not see that. Only some seasonal pattern is stronger in the upper troposphere in ERA5. Can you please explain this more precisely, and/or rephrase the sentence.

- P14L29-30: Halting, complicated sentence, please rephrase.

- P17L6-7: Can you rephrase this sentence please, I think this is important, but I do not understand it.

- P19L1: Why competing? Both these points are accelerating the BDC, right?

- P19L4-5: According to what I can see from the figures, this statement is probably right, but why do you not show the differences figures in that case, as you do for all other figures? (see also my major point on that)

- P20L7: "less evident, but stronger" How can that be? That does not seem right to me. It would be clearer if you would show the differences.

- P20L22: ... can induce meridional wind "changes". (provide a citation for this please, maybe Holton et al. 1995. Also for line 25-27)

- P20L30: this description is not clear to me, please rephrase it.

- P20L31-33: Please rephrase this sentence, it is confusing. Why planetary and gravity wave together in the second part of the sentence? Did you not want to separate the two forcings here?

- P22L4-5: "the difference in the net forcing...." This sentence seems obsolete to me, that is what you analyse here, not a result.

- P22L15-33: Figures that are not shown should not be described in detail and analysed in detail as here. As the paper is quite long already, I do not ask you to show the figures, but rather to move the description and the figures to the supplement and write in the main text only the most important outcome of these figures with a quick reference to the supplement.

- P23L7-8: The trend around the tropopause seems like being caused by the tropopause rise (which is part of stratospheric shrinking). The circulation moves upward. Please reformulate, keep the Sacha citation, but add for example Oberländer-Hayn et al. (2016) and Vallis et al. (2015).

- P23L29: Provide an uncertainty range of the trend (e.g. $1\sigma$)

- P23L23: "not significant", what test did you perform? Significant on what level?

- P23L32-34: please rephrase, it does not disappear, it is simply not present in ERA5, in contrast to ERAi. This is indeed a very important point, because it has been standing in contrast to what GCMs simulate. It should be described/discussed with references in the last section.

- Fig. 11: Panel a: Please do not show these white (undefined) areas in the troposphere, instead, modify the colour bar such that it includes -inf and +inf, so that at least the sign can clearly be seen.
  Panels b-d: What are the contour lines here?
  Add an estimate of the significance of the trends, and the units to the colour bars.

- caption Fig. 12: replace "trend" with "linear regression lines" (both times).

- P26L28: Please provide a citation for ERA5 here.
  Also, remove the last part of the sentence (from ", which is" onward) or clarify what was done there and make it more precise.

- P26L32: But this acceleration has not been observed! Observations show a non-significant deceleration (Engel eta. 2009, 2017). The acceleration has been shown in model simulations.

- P26L33-34: This sentence simplifies the problem too much. Please revise it.

**Technical issues:**

- P1L7: The comparison of .... shows very good....

- P1L9: ...and in the...

- P1L13: ...due to weaker gravity wave forcing at....

- P1L20: ...(BDC, e.g. ...

- P1L21: ... has received a lot of interest ...

- P2L2: remove "most effectively"

- P2L7: (the so-called surf zone)

- P2L16: ... and small-scale waves

- P2L23: greenhouse gas .... ozone deplet**ing** substances

- P2L24-26: ... variability, QBO and ENSO, which affecting the temperature structure and thus the tropical upwelling and extratropical downwelling in the stratosphere

- P2L27: ... in wave propagation...

- P2L31: remove "would"

- P2L32: ...negative stratospheric temperature...

- P4L5and6: change "better" to "higher"

- P4L8: ... for the 2000...

- P4L18: I guess you mean UTC here.

- P4L28: p$_S$ (small letter)

- P4L29: velocity. ...

- P5L3: ... is the Lagrangian...

- P5L17: $\Theta$ is the potential...

- P6L10: ... to maintain steady ...

- P610-11: brackets around citations (use citep)

- P6L17: I guess this should be $\phi(z')$

- P6L24: Used as a ...

- P7L1: ... in detail and applied in numerous...

- P7L8: ... which are the...

- P7L17: ... seasonal means of ...
  This appears also below (P7L28 and caption Fig. 1, maybe more.)

- P7L27: 0.5 mm·s$^{-1}$. (The dot between mm and s!). And that wrong dot appears many times again in the paper, e.g. P7L31, Fig. 2, Fig. 4, please correct throughout the paper

- For a quick look at the figures, I would appreciate the units of the variable attached to each color bar, rather than (only) in the caption.

- caption fig. 1: black

- Fig. 2 d,e,f: Relative "differences" in ...

- Fig. 2 overbar over w

- caption Fig. 2: ... the differences of $\overline{w^*}$ between .... relative to the annual mean.

- caption Fig. 3: Monthly mean $\overline{w^*}$ at 70 hPa as function of latitude and time.

- P10L3: of $\overline{w^*}$

- P10L5: and L16: in $\overline{w^*}$ (please correct that throughout the paper)

- P11L20: ... between 0 and ...

- P11L21: ... and is consistent...

- P11L25: You mean "Fig. 4g-i"

- P11L28: the two reanalysis in Fig. 5,...

- P11L32: ...between ERA5 and... (remove "the" here, and please do that throughout the paper)

- Caption Fig. 4: boreal winter / boreal summer

- Caption Fig. 4: You describe differences in contour lines, but I do not see any difference in the colour bar, so what do you mean?

- Caption Fig. 5: boreal winter / boreal summer

- P14L3: years

- P14L7: average

- P14L8: ...for the full...

- P14L: ... major modes of ...

- P14L15 interannual time-scales

- P15L1: driving of the

- P16L11: i.e.

- P16L15: of the secondary

- Caption Fig. 7. Missing full stop after (10)

- caption Fig. 8: I guess the common abbreviation for standard deviation is "std"

- P18L1: remove "regions of"

- P18L2: while the QBO

- P18L2: remove "again"

- P18L3: altitudes

- P18L9: conditions

- P20L8: stream-function

- P20L13-14: ...with less ...

- P20L15: remove "out"

- P20L19: driving of the

- P20L22: Assuming zonal momentum...

- P20L23: ... the meridional residual wind...

- P22L11:... the non-orographic gravity wave ...

- P22L11: So ERA5 does... please rephrase the sentence

- P22L15: seaonal differences

- P23L9: ...trend provides an...

- P23L10: remove "in ERA5" (that is the case not only in ERA5)

- P23L15: planetary wave breaking

- P23L18: remove "we also note that"

- P23L22: ERA-Interim shows a ...

- P23L32: the

- P25L4: replace "including" with "in particular"

- P25L10: induced modulations

- P25L16: seasonal mean

References:
- Butchart, N., Cionni, I., Eyring, V., Shepherd, T., Waugh, D., Akiyoshi, H., Austin, J., Brühl, C., Chipperfield, M., Cordero, E., et al.: Chemistry-climate model simulations of twenty-first century stratospheric climate and circulation changes, Journal of Climate, 23, 5349- 5374, doi:10.1175/2010JCLI3404.1, 2010.

- Eichinger, R, Šácha, P. Overestimated acceleration of the advective Brewer–Dobson circulation due to stratospheric cooling. QJR Meteorol Soc. 2020; 1-15. https://doi.org/10.1002/qj.3876

- Engel, A., Möbius, T., Bönisch, H., Schmidt, U., Heinz, R., Levin, I., Atlas, E., Aoki, S., Nakazawa, T., Sugawara, S., et al.: Age of stratospheric air unchanged within uncertainties over the past 30 years, Nature Geoscience, 2, 28–31, doi:10.1038/ngeo388, 2009.

- Engel, A., Bönisch, H., Ullrich, M., Sitals, R., Membrive, O., Danis, F., and Crevoisier, C.: Mean age of stratospheric air derived from AirCore observations, Atmospheric Chemistry and Physics, 17, 6825–6838, doi:10.5194/acp-17-6825-2017, 2017.

- Holton, J. R., Haynes, P. H., McIntyre, M. E., Douglass, A. R., Rood, R. B., and Pfister, L. (1995), Stratosphere–troposphere exchange, Rev. Geophys., 33( 4), 403– 439, doi:10.1029/95RG02097.

- Oberländer-Hayn, S., Gerber, E. P., Abalichin, J., Akiyoshi, H., Kerschbaumer, A., Kubin, A., Kunze, M., Langematz, U., Meul, S., Michou, M., Morgenstern, O., and Oman, L. D.: Is the Brewer-Dobson circulation increasing or moving upward?, Geophysical Research Letters, 43, 1772–1779, doi:10.1002/2015GL067545, 2016.

- Vallis, G. K., Zurita, P., Cairns, C., and Kidston, J.: Response of the large-scale structure of the atmosphere to global warming, Quarterly Journal of the Royal Meteorological Society, 141, 1479–1501, doi:10.1002/qj.2456, 2015.

---

## Referee Comment (RC2) · Anonymous Referee #2 · 4 Dec 2020

This is an interesting and overall well written paper comparing the representation of the Brewer-Dobson circulation in the latest ECMWF reanalysis, ERA5, with its predecessor, ERA-Interim. I have only a few general points and some minor revisions to suggest:

General comments:

- While the detailed comparison between ERA5 and ERA-interim is very useful, it would also help to have some discussion linking how these reanalyses compare to other reanalyses. It is briefly mentioned that previous work has shown ERA-interim to be to have too strong upwelling, but it would be nice to know more. Since this paper is part of the S-RIP special issue, it would the authors better tie their work in with the other S-RIP work on the BDC. I'm not sure if the authors are contributors to the S-

[Figure]

RIP BDC chapter, but I would recommend they get in touch with the authors of that chapter and have some content on the broader context of reanalysis representations of the BDC. - I found some of the discussion of the regression modeling confusing, and would appreciate if the authors could make some of this clearer. The most unclear part is the discussion of things like QBO "amplitude variability" that is plotted in figures 7 through 9. This is not really well defined in the paper. I'm guessing it might be the QBO coefficient in the regression fit, or it might be something like the RMS of the QBO timeseries for the fit. Also, the authors regress things like the zonal wind field against zonal wind defined at a specific level (i.e., the QBO defined as zonal wind at 50 hPa), which is a bit odd and requires a bit more nuanced interpretation. When doing something like this, the correlations at a higher level (at, say, 30 hPa) aren't really "caused" by the QBO wind at 50 hPa, but rather reflect the climatological structure of how the equatorial zonal winds propagate downward as part of the QBO. I think the discussion around these figures (7-9) results could be clarified on this point.

Specific comments:

Pg 1, line 21: gain -> gained

Pg 1, line 22: distributions -> distribution

Pg 2, line 6: ascent -> ascend

Pg 2, lines 31-32: "... strengthen the BDC, consistent with negative temperature trends" – Do you mean negative temperature trends in the tropical lower stratosphere? Please clarify this statement

Pg 2, line 34: e.i. -> i.e.

Pg 2-3: I'm a bit confused by the statement that reanalyses show BDC lower branch strengthening whereas figures like Fig 12 seem to indicate a slowdown (at least in ERA-interim). Are the reanalyses actually consistent on this, or is it just some of them? This comment ties in with my general comment about placing the ECMWF results within the

broader context.

Pg 4: Both ERA5 and ERA-Interim use a hybrid vertical coordinate system. Are the authors using model level data for both calculations, or the lower vertical resolution pressure-gridded data typically provided by reanalysis centers? If using model level data, please state so explicitly. Also, it would be helpful to have some mention what kind of error in the TEM calculations would be induced by using data that is not strictly on pressure levels. Related to this, what is the lowest level (i.e., highest pressure value) at which the model levels are pure pressure levels (i.e., where b=0 for a vertical coordinate system where p = a + b*PS), and is this different between ERA-Interim and ERA5? This might impact the discussion of the residual circulation below 20 km.

Pg 4, line 19: What do the authors mean as the ERA5 dynamical fields are archived as "tendencies" over one hour? I assume fields like u,v,T are either archived as averages over some period (e.g., 1 hour) or are instantaneous quantities.

Pg 4, line 29: Omega is introduced here, but is not used until f is defined in the following sentence. I recommend to define omega in the sentence defining f.

Pg 5, line 3: "is a the" -> "is a"

Pg 6, line 10-11: references need proper parentheses.

Pg 7, line 8: "which the" -> "which are the"

Pg 9, Fig 2: I believe the lower row of plots is showing ERA5 minus ERA-Interim, expressed as a percent difference. Please make this more clear in the caption.

Pg 10, line 9: I don't really see ERA5 being stronger than ERA-Interim above 20 km in general. It does appear to be stronger from about 35 to 40 km (Fig 2e).

Pg 10, line 11: The use of a 15 sigma error bar strikes me as odd. Please justify the use of this quantity.

Pg 10, line 11-12: I don't see what the authors are pointing out here. It looks to me like

the variance increases with height.

Pg 14, line 7-8: Why is only a 2 year period used? Regarding the period used, I assume the 2017-2018 period is somewhat of a best case scenario given that there are more/better observations constraining the reanalysis in this period than, say, in the earliest part of the record. Pg 14, line 14: mode -> modes

Pg 14, line 32: For those of us not intimately familiar with non-orographic gravity wave drag parameterizations, could the authors give some brief information on the difference between the schemes?

Pg 17, line 10: Fig 8c,d should be Fig8 a,b.

Pg 18, line 5: Fig 8a,b should be Fig8 c,d.

Pg 22, line 33: awkward wording. Maybe the authors mean "therefore" instead of "there"?

Pg 24, Fig 11a. I don't understand why the authors are showing the streamfunciton here instead of w*

Pg 24, line 3 and Fig 12: The authors should show the uncertainties on these trends. Does the ERA5 trend use the ERA5 v5.1 data for the period 2002-2007? It would be really helpful to include both ERA5 and 5.1 in the timeseries plot, trend numbers, and discussion. It may not matter, but it's not obvious whether or not that is the case.

Pg 26, line 22: Could the authors be more specific about resolved vs. unresolved GW forcing in the discussion here, and elsewhere in the manuscript?

Pg 26, line 24: large -> larger

---

## Author Response (AR1)

**Answer to Roland Eichinger's comments on "The advective Brewer-Dobson circulation in the ERA5 reanalysis: climatology, variability and trends" by Mohamadou Diallo et al.**

Dear Editor-in-Chief, Peter H. Haynes,

We are submitting our revised article titled "'The advective Brewer-Dobson circulation in the ERA5 reanalysis: climatology, variability and trends". We thank the two Reviewers for their detailed and well thought-out comments, which helped to significantly improve the paper. We have made substantial changes to the manuscript in order to thoroughly address the Reviewers' suggestions and comments. Main changes concern:

- The calculation of residual circulation from wave drag using the downward control principle, as suggested by Reviewer #1, a new figure showing these results and the related discussion.

- Addition of statistical significance using Student's t-test to the differences as suggested by Reviewer #1

- Addition of information related to S-RIP and references.

- Re-calculation of the RCTT using the w* instead of heating rates for 2010-2018.

- rephrasing of several paragraphs in order to clarify the manuscript.

With these changes, we are convinced that the paper has been significantly improved and is highly relevant for a wide-ranging journal like *Atmospheric Chemistry and Physics*. Please see below our answers point by point to all reviewers comments and suggestions.

Reviewers comments are in bold, followed by our respective replies. Changes in the manuscript are in blue, allowing them to be tracked easily.
Kind regards,
Mohamadou Diallo (on behalf of the co-authors)

**Roland Eichinger, Reviewer #1 (Comments to Author):**

**Major issues:**

1. ***P1L12-13 and P25L12-13: I did not believe this statement when I first read the abstract and I still dont do so having read the whole paper. Commonly the contribution of gravity waves on tropical upwelling is around 30% here (see Butchart et al. 2010). Stating that a weaker GW forcing reduces tropical upwelling by 40% does not go together with that. The statement seems to base on the sentence "The contribution of the planetary waves to the tropical upwelling differences is less evident" on P21L1-2 (and P26L23), which you use to entirely disregard any PW contribution or anything else. I think what could help to separate the contributions of planetary and GW waves here could be a downward control (Haynes et al. 1991) analysis, but on the basis of patchy Fig. 10, the statement seems adventurous to me. Moreover, how well do the tropopauses fit together between ERAi and ERA5? The differences seem strongly altitude-dependent. This and also Fig. 2b made me think of a possible vertical shift between the reanalyses, that could contribute to the upwelling differences, too.***

    Thank you Roland for this comment and suggestion, which we followed further with carrying out the suggested downward control analysis, what significantly improved the paper! First of all, we agree that this strong contribution due to GWs seems somewhat inconsistent with Butchart et al 2010. However, the small-scale GWs depend largely on the models resolution and parameterizations. As both (resolution, GW parameterization) clearly differ between ERA5 and ERA-Interim, a significant effect on the reanalysis GWD can be expected. Furthermore, it is important to notice (Fig. 2b) that the 40% difference in the upwelling occurs only in a shallow layer between 15-17km, above the difference decreases with height.

    The new statistical significance estimate Fig. 10 shows that the differences in GWD between the reanalysis in the lower stratosphere subtropics are indeed significant (at 95% level). Finally, our downward calculation indeed quantitatively proved that the difference in the upwelling between ERA5 and ERA-Interim is related to GW drag. The fact that PWD agrees much better between ERA5 and ERA-Interim

makes sense to us because PWs are well resolved now in the reanalyses and climate models, therefore, differences between the reanalyses should only to a minor extent be caused by differences in the representation of PWs.

Regarding the potential effect of a tropopause difference between the reanalyses, we think that such an effect should be minor. The tropopause between the two reanalyses has already been compared by Tegtmeier et al (2020) that we already cited in the manuscript. They concluded that "there is good agreement between reanalysis estimates of tropical mean temperatures and radio occultation data, with relatively small cold biases for most data sets". Thanks again for this very good suggestion!

2. ***In all plots where you calculate the difference between ERA-interim and ERA5 you need to add the statistical significance of the differences. In some regions and/or for some variables, the variability may be so large that the mean differences are not meaningful or small differences might be overseen despite their significance. Please add significance to all those plots and revise your text accordingly. In many occasions (e.g. P26 L16,L24,31) you even mention significance, but it has never been shown.***

Thank you Roland for this remark. We have added the statistical significance (t-test) of the differences in the figures and marked where results are significant. We revised the text where required.

3. ***P7L26-27 and P25L13-14: "suggesting a stronger advective BDC" That statement seems much too general and in some sense even wrong, because you already showed, that in the tropics, the upwelling is weaker. Often tropical upwelling is used as a proxy for the advective BDC in the whole stratosphere. As the stronger downwelling is limited to the high latitudes, this might be related to the polar vortex strength, which has some impact on high latitude downwelling. But the polar vortex differences between ERAi and ERA5 are not discussed in the paper. I am not saying you should analyse it, and include plots, but there should be a discussion about it, separating the different regions. The topic only appears very briefly in the very last sentence of the paper. In my opinion this is much too late and too briefly discussed, it should be included properly in the analysis and discussion.***

We agree that the formulation here was misleading. The upwelling is weaker compared to the fast ERA-I upwelling, but at the same time ERA5 does show a stronger downwelling of advective BDC in polar regions (Fig 1 and 2). The effect of polar vortex strength could indeed affect the downwelling differences, but would need a more detailed analysis beyond the scope of this paper. What is sure is that several things might be in play such as a higher resolution, higher model top, new non-orographic gravity wave parameterization scheme, and the progress made in the more realistic representation of the sponge layer in ERA5, which is presumably designed to damp upward propagating waves near the model top to avoid reflections from the upper boundary. Thus, we modified the text to clarify, and furthermore report what the analyses show and avoid any speculation about the polar vortex without background information. Further, we have improved the discussion.

4. ***The paper is quite long and not every analysis is really needed for the conclusions. I think some of the figures (or panels) do not contribute anything to the final conclusions. I therefore suggest to go through the paper starting from the back, analysing each conclusion with respect to which figures are really needed for that. The rest can be banished to a supplement, where I would also move the figures that now are in the appendix. For example: P25L14-18: If this is all the outcome from the seasonal climatologies, I think the analysis of the seasonal climatologies can be reduced drastically, and most of the figures can be moved to a supplement. Also, I am not sure if Fig. 3 is needed at all for the conclusions.***

We agree that the paper is a bit long. However, as the ERA5 Thank you Roland for the suggestion. As the ERA5 reanalysis is new and this manuscript is the first paper looking at metrics of BDC for the S-RIP, we would like to present a complete analysis on the advective BDC representation in ERA5. Hence, we decided to keep all figures. However, we went through the text to streamline and shorten it, where possible.

5. ***Until section 4, the paper always focuses on the comparison between ERAi and ERA5. In section 4, at first only the ERA5 trends (streamfct and wave drag trends) are analysed, and then the mass flux trend is compared between ERA5 and ERAi again. I do not understand why you do not stick to the comparison between the reanalysis products, as it seemed to me that this is the focus of the paper. I suggest to include the ERA5-ERAi streamfct and wave drag comparison here as well, to keep the theme of the paper. Moreover, I suggest to reflect that theme also in the title of the***

*paper, maybe something like: "Comparison of the advective Brewer-Dobson circulation between ERA5 and ERAinterim reanalysis: climatology, variability and trends". And I particularly suggest to add "climatology", because that is what most of the paper deals with (annual and seasonal climatologies).*

We do see the point that Fig. 11 is a bit different to the style of the other figures. However, as many authors have already discussed the effects of different wave types on BDC trends in ERA-Interim (e.g., Abalos et al., 2015) and repeating this here would add another 4 plot panels we decide to keep the figure as is. Nevertheless, we have improved the discussion and links toward previous studies. Also, we have added the term "climatology" into the title.

6. *The discussion and conclusion section almost completely lacks the connection to literature and it also does not point out what the implications of the study are. I think both these points can easily be addressed. The literature is already outlined nicely in the introduction. The implications can include what the present study means for older conclusions about the BDC based on ERAi data (or model simulation results that are nudged or prescribed to ERAi dynamics), and what now can be assessed better, or more precisely, or differently with ERA5. This would be nice for closing remarks of the paper.*

Thanks for this opinion! As we do not want to extend the conclusion section too much, we decided to address these points in section 4, which we modified accordingly. Now, we discuss the implications of our results for climate model simulations that are nudged to reanalysis meteorology and using reanalysis for validation.

**Minor issues:**

1. *P3L11: "the strength of the BDC". Do you mean the change of the strength here or really the strength? If the latter, I think the statement is out of place here, since you talk about trends in this paragraph.*

We mean both: the strength and the trend. This is now stated more clearly in the revised manuscript.

2. *P3L13: remove "reanalyses and for". There were no reanalyses analysed in that study.*

We have rephrased the sentence.

3. *P4L11-12: I assume that the ability to better resolve (macro-scale) meteorological features is mainly due to the higher resolution, and not "apart from that". Hence, please reformulate the sentence.*

The sentence is talking about the improvement in ERA5 compared to ERA-Interim. In addition to higher resolution, there are additional important improvements as we mentioned in this paragraph. We have clarified it.

4. *Sect. 2.2: Please consider the QJRMS paper by Eichinger and Sacha, (2020). You have to be very cautious with which w* you use for your analyses. As you do all your calculations in log-pressure coordinates, you may make a mistake when you calculate the w* trends because in log-pressure coordinates, the trends of the pressure levels (i.e. the temperature trends and thereby the trend in the scale-height, which you implicitly neglect using the log-pressure formulae) is not considered. In that paper, we propose using the goepotential coordinates. However, for your streamfunction trends, you should be fine, because the H cancels out when you use the log-pressure density (Sect. 4.2 of the paper). But you also calculate some w* trends at the end where it certainly matters. And I am not exactly sure if it is important for the variability calculations. Please make sure your analyses do not include the error.*

We are aware of this study and conclusions. As both estimates are consistently calculated (e.i. literally saying Apple to Apple comparison). Any bias in one will be present in the other one. So multiplying our result by a constant (20%) will just shift the value but won't changes any pattern and variability differences between ERA5 and ERA-Interim. In addition, we have done the calculation on the original model levels for enabling comparisons to previous results that were obtained for reanalysis comparison S-RIP project as this article is part of the project.

5. *P7L4: Remove the overbar over w*, it is not zonal mean here. But throughout the paper, use the overbar when zonal mean is shown and no overbar if it is general. And note somewhere that the overbar stands for zonal mean.*

In the regression, we use also the zonal mean w*bar as defined in equation (3).

6. ***Fig. 2: Why 15.$\sigma$? The 15 seems very arbitrary, can you explain why you chose it? And why not adding such a measure of uncertainty also to a,b,c? It would be helpful when interpreting the results (see also my major point about statistical significance)***

   The 15.$\sigma$ was just for scaling the figure as the vertical profile are in percentage and sigma not. Anyway, we have replace the errorbar by the statistically significant value.

7. ***P10L9: But according to Fig. 2e, that is only true between 35 and 40 km, not everywhere above 20 km.***

   We have rephrased the sentence and added new ones.

8. ***P11L14-16: Note that ... polar regions. I do not understand what you try to say with that sentence. I suggest to remove it.***

   We are saying that there are differences in the upwelling and downwelling region like tropics, surf-zone and polar. We have rephrased it.

9. ***P11L19-24: The description of the differences is unclear to me: - In DJF, I only see a small negative patch in the UTLS and more negative fields in the SH mid to high latitudes. The conclusion in L21 therefore bases on wrong facts. - I also do not see the positive differences between 40S and 0, where are you there? Still in DJF? The differences generally barely exceed the tropopause, or are just much smaller in the stratosphere, as the mass streamfunction is much smaller there too. So the color bar is maybe not really suitable, and moreover, it would be very helpful here to see statistical significance. The statement that the residual circulation is stronger in ERA5 seems wrong. Firstly, you already showed that upwelling is weaker, secondly Fig. 4c shows the opposite. Please clarify and correct these points.***

   Thank you for this comments. Our apologies! We had made a mistake in indicating the right figure. The Fig. 4f and 4i clearly shows a blue area (negative difference) extending from the UTLS into the upper stratosphere during the DJF. Regarding the statistical significance, we have added and included the discussion in the revised version where it's needed.

10. ***P11L25-27: I do not understand what you mean by "horizontal shift" here. Can you elaborate on this please.***

    What we mean is that the circulation cell look like being southward shift in the UTLS during winter and northward shift during summer compared to ERA-Interim. We have rephrased it.

11. ***P14L23: "stronger westerly and easterly shear" I can not quite see that. Can you please describe where/guide the eye a bit?***

    We drop the term "shear". ERA5 exhibits stronger westerly and easterly QBO winds compared to the ERA-Interim as shown in the Fig. 6c (cross section) in the whole topical region. We have rephrased it.

12. ***P14L25: "into the troposphere below the tropopause .... does not propagate that far downward" I can not see that. Only some seasonal pattern is stronger in the upper troposphere in ERA5. Can you please explain this more precisely, and/or rephrase the sentence.***

    We have rephrased it.

13. ***P14L29-30: Halting, complicated sentence, please rephrase.***

    We have rephrase it as following "In the ERA5, the use of a new non-orographic gravity wave drag parameterization, which is different from the ERA-Interim, is likely the cause of the observed differences (Orr et al., 2010; Scaife et al., 2002; Lott et al., 2012; Richter et al., 2014)."

14. ***P17L6-7: Can you rephrase this sentence please, I think this is important, but I do not understand it.***

    We have rephrased it.

15. ***P19L1: Why competing? Both these points are accelerating the BDC, right?***

    Competing in the sense which one has a predominant effect for instance on mid-latitude ozone, i.e. global warming versus strong El niño. However in term of acceleration, they both led to BDC strengthening. We have rephrased it and be more explicit.

16. **P19L4-5: According to what I can see from the figures, this statement is probably right, but why do you not show the differences figures in that case, as you do for all other figures? (see also my major point on that)**

    Thank you for the suggestion. As the difference is clearly visible, therefore, we don't think it is necessary showing an additional panel.

17. **P20L7: "less evident, but stronger" How can that be? That does not seem right to me. It would be clearer if you would show the differences.**

    We mean that the ENSO-induced change in the deep branch is much stronger in ERA5 than in ERA-i even though they are not as clear as in the shallow and transition branches. We rephrased the sentence.

18. **P20L22: ... can induce meridional wind "changes". (provide a citation for this please, maybe Holton et al. 1995. Also for line 25-27)**

    In this sentence "contribution" and "changes" are inter-changeable. We keep it. We have added the citation.

19. **P20L30: this description is not clear to me, please rephrase it.**

    We have rephrase it.

20. **P20L31-33: Please rephrase this sentence, it is confusing. Why planetary and gravity wave together in the second part of the sentence? Did you not want to separate the two forcings here?**

    What we mean is "in the UTLS GW drag is dominant while in the upper stratosphere both GW and PW drag contribute. We have rephrased it.

21. **P22L4-5: "the difference in the net forcing..." This sentence seems obsolete to me, that is what you analyse here, not a result.**

    The net forcing is shown in fig 10 and is calculated as (PW+GW-dudt). We have rephrased it.

22. **P22L15-33: Figures that are not shown should not be described in detail and analysed in detail as here. As the paper is quite long already, I do not ask you to show the figures, but rather to move the description and the figures to the supplement and write in the main text only the most important outcome of these figures with a quick reference to the supplement.**

    Due to the importance of the figure and the discussion, we have chosen this compromised solution. Therefore, the discussion will stand for itself. Reference to the figures in the Appendix is only given for completeness.

23. **P23L7-8: The trend around the tropopause seems like being caused by the tropopause rise (which is part of stratospheric shrinking). The circulation moves upward. Please reformulate, keep the Sacha citation, but add for example Oberlnder-Hayn et al. (2016) and Vallis et al. (2015).**

    We have added the references, as suggested. However, we would like to keep the word "probably", as further investigation of the strength of this process is still needed for ERA5. We have added the references and rephrased it.

24. **P23L29: Provide an uncertainty range of the trend (e.g. $1\sigma$)**

    We have include uncertainty range.

25. **P23L23: "not significant", what test did you perform? Significant on what level?**

    We use a student t-test with two tail distribution and the level of significant is 95%. We have added in the text.

26. **P23L32-34: please rephrase, it does not disappear, it is simply not present in ERA5, in contrast to ERAi. This is indeed a very important point, because it has been standing in contrast to what GCMs simulate. It should be described/ discussed with references in the last section.**

    We have rephrased it and added references.

27. ***Fig. 11: Panel a: Please do not show these white (undefined) areas in the troposphere, instead, modify the colour bar such that it includes -inf and +inf, so that at least the sign can clearly be seen. Panels b-d: What are the contour lines here? Add an estimate of the significance of the trends, and the units to the colour bars.***

   Thank you for the suggestion. We have included these white (undefined) areas in the troposphere. We did not add statistical significant values as it is not significant at 95%. It does not make sense. The contour lines are zonal wind and we have added to the captions. For aesthetical reasons of the figures, we just put the units in the captions.

28. ***caption Fig. 12: replace "trend" with "linear regression lines" (both times).***

   For consistency with the discussion in our regression section, and to be more accurate, we replaced "trend" with "linear trend".

29. ***P26L28: Please provide a citation for ERA5 here. Also, remove the last part of the sentence (from ", which is" onward) or clarify what was done there and make it more precise***

   We have added citations.

30. ***P26L32: But this acceleration has not been observed! Observations show a non-significant deceleration (Engel eta. 2009, 2017). The acceleration has been shown in model simulations.***

   Here, we were explained the observed temperature trends in the lower stratosphere from the findings of Thompson and Solomon, 2005 as well Fu and Qu, 2014 all agreed in the acceleration of the BDC upwelling. The Engel et al 2017 only look at a very localized area in the NH upper stratosphere, which is remote from the tropical upwelling and not significant. The previous papers look at observations in the most relevant area, including the tropical upwelling region. We have precise what we mean by rephrasing the sentence.

**Minor issues:**

1. ***P1L7: The comparison of .... shows very good....***

   We have rephrased it.

2. ***P1L9: ...and in the...***

   Our original wording and that suggested are both correct. We decided to keep the wording as is.

3. ***P1L13: ...due to weaker gravity wave forcing at....***

   We have rephrased it.

4. ***P1L20: ...(BDC, e.g. ...***

   We have rephrased it.

5. ***P1L21: ... has received a lot of interest ...***

   We have rephrased it

6. ***P2L2: remove "most effectively"***

   We decided to keep the wording as is.

7. ***P2L7: (the so-called surf zone)***

   We decided to keep the wording as is.

8. ***P2L16: ... and small-scale waves***

   We rephrased it.

9. ***P2L23: greenhouse gas .... ozone depleting substances***

   We rephrased it.

10. ***P2L24-26: ... variability, QBO and ENSO, which affecting the temperature structure and thus the tropical upwelling and extratropical downwelling in the stratosphere***

   We decided to keep the wording as is.

11. ***P2L27: ... in wave propagation...***

We have rephrased it.

12. ***P2L31: remove "would".***

We have rephrased it.

13. ***P2L32: ...negative stratospheric temperature...***

We have rephrased it to the suggested text by Reviewer 2.

14. ***P4L5and6: change better to higher***

We decided to keep as it is because we mean better as "higher" is already illustrated by brackets "(6-hourly versus 1-5hourly)".

15. ***P4L8: ... for the 2000...***

We have rephrased it.

16. ***P4L18: I guess you mean UTC here.***

We have rephrased it.

17. ***P4L28: ps (small letter)***

We have rephrased it.

18. ***P4L29: velocity***

We have rephrased it.

19. ***P5L3: ... is the Lagrangian...***

We rephrased it following Reviewer 2 suggestion.

20. ***P5L17: $\theta$ is the potential...***

We have rephrased it.

21. ***P6L10: ... to maintain steady ...***

We decided to keep the wording as is.

22. ***P6L10-11: brackets around citations (use citep)***

We have rephrased it.

23. ***P6L17: I guess this should be $\phi(z')$***

It is $\phi(z)$ as z' is just a substitute to z for notation sake in the integral. We decided to keep the wording as is.

24. ***P6L24: Used as a ...***

We have rephrased it.

25. ***P7L1: ... in detail and applied in numerous...***

We have rephrased it

26. ***P7L8: ... which are the...***

We have rephrased it.

27. ***P7L17: ... seasonal means of ... This appears also below (P7L28 and caption Fig. 1, maybe more.)***

We decided to keep the wording as is.

28. **P7L27: 0.5 mm.s1. (The dot between mm and s!). And that wrong dot appears many times again in the paper, e.g. P7L31, Fig. 2, Fig. 4, please correct throughout the paper**

    We have replaced the dot by the common way ("cdot").

29. **For a quick look at the figures, I would appreciate the units of the variable attached to each color bar, rather than (only) in the caption.**

    Instead of overloading the plots with information we decided to state the units in the captions.

30. **caption fig. 1: black**

    We have corrected it.

31. **Fig. 2 overbar over w**

    With the matlab version that we have, it is not possible to put the overbar.

32. **caption Fig. 2: ... the differences of w* between .... relative to the annual mean**

    We have rephrased it.

33. **caption Fig. 3: Monthly mean w* at 70 hPa as function of latitude and time.**

    We decided to keep the wording as is as "Monthly" and "latitudinal" are equivalent to "as a function of latitude and time"

34. **P10L3: of w***

    We decided to keep the wording as is.

35. **P10L5: and L16: in w* (please correct that throughout the paper)**

    We decided to keep the wording as is.

36. **P11L20: ... between 0 and ...**

    We have rephrased it.

37. **P11L21: ... and is consistent...**

    We have rephrased it.

38. **P11L25: You mean "Fig. 4g-i"**

    We have rephrased it.

39. **P11L28: the two reanalysis in Fig. 5,...**

    We decided to keep the wording as is.

40. **P11L32: ...between ERA5 and... (remove the here, and please do that throughout the paper)**

    We have rephrased it.

41. **Caption Fig. 4: boreal winter / boreal summer**

    We have rephrased it.

42. *Caption Fig. 4: You describe differences in contour lines, but I do not see any difference in the colour bar, so what do you mean?*

We decided to keep the wording as is.

43. *Caption Fig. 5: boreal winter / boreal summer*

We have rephrased it.

44. *P14L3: years*

We have rephrased it.

45. *P14L7: average*

We have rephrased it.

46. *P14L8: ...for the full..*

We decided to keep the wording as is.

47. *P14L: ... major modes of ...*

We have rephrased it.

48. *P14L15 interannual time-scales*

We have rephrased it.

49. *P15L1: driving of the*

We have rephrased it.

50. *P16L11: i.e.*

We have corrected it.

51. *P16L15: of the secondary*

We have rephrased it.

52. *Caption Fig. 7. Missing full stop after (10)*

We have corrected it

53. *caption Fig. 8: I guess the common abbreviation for standard deviation is std*

We have corrected it as ACP imposed this abbreviation.

54. *P18L1: remove "regions of"*

We have rephrased it

55. *P18L2: while the QBO*

We have rephrased it.

56. **P18L2: remove "again"**
We have deleted it

57. **P18L3: altitudes**
We have rephrased it.

58. **P18L9: conditions**
We have rephrased it.

59. **P20L8: stream-function**
We have rephrased it.

60. **P20L13-14: ...with less ...**
We have rephrased it.

61. **P20L15: remove out**
We have rephrased it.

62. **P20L19: driving of the**
We have rephrased it.

63. **Assuming zonal momentum...**
We decided to keep the wording as is.

64. **P20L23: ... the meridional residual wind...**
We decided to keep the wording as is.

65. **P22L11:... the non-orographic gravity wave ...**
We have rephrased it.

66. **P22L11: So ERA5 does... please rephrase the sentence**
We have rephrased it.

67. **P22L15: seaonal differences**
We decided to keep the wording as is.

68. **P23L9: ...trend provides an...**
We have rephrased it.

69. **P23L10: remove in ERA5 (that is the case not only in ERA5)**
We are talking about our result. We decided to keep the wording as is.

70. ***P23L15: planetary wave breaking***

    We have rephrased it.

71. ***P23L18: remove we also note that***

    We have rephrased it.

72. ***P23L22: ERA-Interim shows a ...***

    We have rephrased it.

73. ***P23L32: the***

    Does not apply anymore. This text was generally revised.

74. ***P25L4: replace including with in particular***

    We have rephrased it.

75. ***P25L10: induced modulations***

    Does not apply anymore. This text was generally revised.

76. ***P25L16: seasonal mean***

    We decided to keep the wording as is.

**Anonymous Referee #2:**

**General comments:**

1. ***- While the detailed comparison between ERA5 and ERA-interim is very useful, it would also help to have some discussion linking how these reanalyses compare to other reanalyses. It is briefly mentioned that previous work has shown ERA-interim to be to have too strong upwelling, but it would be nice to know more. Since this paper is part of the S-RIP special issue, it would the authors better tie their work in with the other S-RIP work on the BDC. I'm not sure if the authors are contributors to the S-RIP BDC chapter, but I would recommend they get in touch with the authors of that chapter and have some content on the broader context of reanalysis representations of the BDC.***

    We thank the Reviewer for this thoughtful suggestions. We are involved in several S-RIP chapter, including the S-RIP BDC chapter 5. We have enhanced the discussion about the comparison to other reanalyses, about the too strong upwelling in ERA-Interim and the S-RIP work on the BDC at needed places in the manuscript (page 4, lines 26-30 and section 4).

2. ***- I found some of the discussion of the regression modeling confusing, and would appreciate if the authors could make some of this clearer. The most unclear part is the discussion of things like QBO amplitude variability that is plotted in figures 7 through 9. This is not really well defined in the paper. I'm guessing it might be the QBO coefficient in the regression fit, or it might be something like the RMS of the QBO timeseries for the fit. Also, the authors regress things like the zonal wind field against zonal wind defined at a specific level (i.e., the QBO defined as zonal wind at 50 hPa), which is a bit odd and requires a bit more nuanced interpretation. When doing something like this, the correlations at a higher level (at, say, 30 hPa) aren't really "caused" by the QBO wind at 50 hPa, but rather reflect the climatological structure of how the equatorial zonal winds propagate downward as part of the QBO. I think the discussion around these figures (7-9) results could be clarified on this point.***

    We thanks the Reviewer for pointing this out. The simplified description of the regression model was motivated by the fact that the regression model is well described in our previous studies (e.g. Diallo et al.

2018, 2019). Our regression model uses a lag term, therefore, allowing us to use only one QBO proxy at any given level. With the lag term, we can reconstruct the propagation of the QBO signal, therefore preventing us using two QBO proxies as classical linear regressions do. Regarding the figures 7 through 9, the Reviewer is right as the QBO "amplitude variability" is the QBO coefficient in the regression fit normalized by the standard deviation of the QBO proxy for all plots except for the temperatures and zonal mean wind figure. Now, we have clarified the description of the regression model and enhanced the discussion of the regression results.

**Specific comments:**

1. ***Pg 1, line 21: gain $->$ gained***

   We have rephrased it.

2. ***Pg 1, line 22: distributions $->$ distribution***

   We have rephrased it.

3. ***Pg 2, line 6: ascent $->$ ascend***

   We have rephrased it.

4. ***Pg 2, lines 31-32: "strengthen the BDC, consistent with negative temperature trends" Do you mean negative temperature trends in the tropical lower stratosphere? Please clarify this statement***

   We thank the Reviewer for precision. Yes indeed, we mean the negative temperature trends in the tropical lower stratosphere as shown in previous studies (Thompson and Solomon, 2005; Fu et al., 2019). We have rephrased the sentences.

5. ***Pg 2, line 34: e.i. $->$ i.e.***

   We have rephrased it.

6. ***Pg 2-3: I'm a bit confused by the statement that reanalyses show BDC lower branch strengthening whereas figures like Fig 12 seem to indicate a slowdown (at least in ERAinterim). Are the reanalyses actually consistent on this, or is it just some of them? This comment ties in with my general comment about placing the ECMWF results within the broader context.***

   We apologize for the confusion. Overall, all reanalysis data sets except CFSR agreed in a robust strengthening shallow branch of the BDC from several different diagnostics such age of air, age spectrum and TEM (Diallo et al. 2012; Abalos et al. 2015; Chabrillat et al. 2018; Ploeger et al. 2019). The negative trend in ERA-interim w* calculated using the standard TEM formula is only present in the ERA-Interim as we have mentioned it in the manuscript following the previous studies (Seviour et al 2011; Abalos et al 2015). Abalos et al 2015 using different ways (standard, momentum and thermodynamic balance) of estimating w*, show that only the standard formula exhibits that behavior. In figure 12, we re-evaluated and highlighted that this strange behavior in ERA-interim w* calculated using the standard TEM formula vanishes in ERA5. Thus, this allows our scientific community to use ERA5 w* estimated with the standard TEM formula as a proxy for climate model inter-comparison, which were not recommended for ERA-Interim w* estimated using the standard TEM formula.

7. ***Pg 4: Both ERA5 and ERA-Interim use a hybrid vertical coordinate system. Are the authors using model level data for both calculations, or the lower vertical resolution pressure-gridded data typically provided by reanalysis centers? If using model level data, please state so explicitly. Also, it would be helpful to have some mention what kind of error in the TEM calculations would be induced by using data that is not strictly on pressure levels. Related to this, what is the lowest level (i.e., highest pressure value) at which the model levels are pure pressure levels (i.e., where b=0 for a vertical coordinate system where p = a + b\*PS), and is this different between ERA-Interim and ERA5? This might impact the discussion of the residual circulation below 20 km.***

   Thank you for for these comments and suggestions about the lowest level (b=0). We use model level data interpolated to log-pressure levels for both calculations. Fortunately, the ERA5 and ERA-interim shows almost the same lowest level (i.e., highest pressure value) at which the model levels are pure pressure levels (i.e., where b=0 for a vertical coordinate system where p = a + b\*PS). This level is about 18 km. For ERA-Interim, the coefficient b is equal to zero where the coefficient a is equal to 7306.63000 Pa and for ERA5 b is equal to zero where a is equal to 7311.869141 Pa.

8. ***Pg 4, line 19: What do the authors mean as the ERA5 dynamical fields are archived as tendencies over one hour? I assume fields like u,v,T are either archived as averages over some period (e.g., 1 hour) or are instantaneous quantities.***

   We have rephrase it.

9. ***Pg 4, line 29: Omega is introduced here, but is not used until f is defined in the following sentence. I recommend to define omega in the sentence defining f.***

   We have moved the definition of omega at suggested place.

10. ***Pg 5, line 3: is a the $->$ is a***

    We have rephrased it.

11. ***Pg 6, line 10-11: references need proper parentheses.***

    We have rephrased it.

12. ***Pg 7, line 8: "which the" $->$ "which are the"***

    We have rephrased it.

13. ***Pg 9, Fig 2: I believe the lower row of plots is showing ERA5 minus ERA-Interim, expressed as a percent difference. Please make this more clear in the caption.***

    Yes indeed, the ERA5 minus ERA-Interim, expressed as a percent difference is actually the relative difference in percentage. We have rephrased it.

14. ***Pg 10, line 9: I don't really see ERA5 being stronger than ERA-Interim above 20 km in general. It does appear to be stronger from about 35 to 40 km (Fig. 2e).***

    Yes, that's correct. We have corrected it.

15. ***Pg 10, line 11: The use of a 15 sigma error bar strikes me as odd. Please justify the use of this quantity.***

    This use of 15 sigma was just for scaling issues as the differences were plotted in percentage. We have now added as errorbar the statistical significance of the differences, which highlight better the areas where the differences between the two reanalyses are significant.

16. ***Pg 10, line 11-12: I don't see what the authors are pointing out here. It looks to me like the variance increases with height.***

    We have rephrased it.

17. ***Pg 14, line 7-8: Why is only a 2 year period used? Regarding the period used, I assume the 2017-2018 period is somewhat of a best case scenario given that there are more/better observations constraining the reanalysis in this period than, say, in the earliest part of the record. Pg 14, line 14: mode $->$ modes.***

    We used two 2 years of mean RCTT because that's what we had at that time. Now, we replaced the RCTT calculated from diabatic heating rates for these 2 years by RCTTs calculated from (v*,w*) TEM residual circulation over the much longer period 2010-2018. The result is still consistent with fact ERA5 is be slower than ERA-interim.. The respective text passages are changed accordingly.

18. ***Pg 14, line 32: For those of us not intimately familiar with non-orographic gravity wave drag parameterizations, could the authors give some brief information on the difference between the schemes?***

    We have added brief information about the schemes.

19. ***Pg 17, line 10: Fig 8c,d should be Fig8 a,b.***

    We have corrected it.

20. ***Pg 18, line 5: Fig 8a,b should be Fig8 c,d.***

    We have corrected it.

21. ***Pg 22, line 33: awkward wording. Maybe the authors mean "therefore" instead of "there"?***

    We have rephrased it.

22. ***Pg 24, Fig 11a. I dont understand why the authors are showing the streamfunciton here instead of w\****

   The mass streamfunction and w* are equivalent. The advantage using the streamfunction is that we can distinguish the BDC cells, and therefore discuss the different BDC branches (e.g., shallow vs. deep branch).

23. ***Pg 24, line 3 and Fig 12: The authors should show the uncertainties on these trends. Does the ERA5 trend use the ERA5 v5.1 data for the period 2002-2007? It would be really helpful to include both ERA5 and 5.1 in the timeseries plot, trend numbers, and discussion. It may not matter, but it's not obvious whether or not that is the case.***

   We have added the uncertainties to these trends. Yes, we use the ERA5 v5.1 in the study as we state in the section of description of the reanalyses.

24. ***Pg 26, line 22: Could the authors be more specific about resolved vs. unresolved GW forcing in the discussion here, and elsewhere in the manuscript?***

   We have rephrased it.

25. ***Pg 26, line 24: large − > larger***

   We have rephrased it.

---

## Referee Report (RR1)

**Review of the revised version of:**
**"The advective Brewer-Dobson circulation in the ERA5 reanalysis: climatology, variability and trends", ACPD, 2020, by Diallo, M. et al..**

The authors have taken into account all comments from both reviewers, or have stated good reasons for not considering some of them. I think the paper is now good and can be published in ACP. Rereading it, I still stumbled across a number of minor or technical issues that can be corrected without problems, I list them below. Particularly in the sections 3.4, 4 and 5, there are a couple of inaccurate statements that should be handled with care, though.

Thanks for the nice study and best wishes
Roland

Note that line numbers refer to the version with the blue highlighted track changes.

**Minor issues:**

- P1L4: ...inter-annual variability, climatology and trends...

- P1L4: ...with the predecessor ERA-Interim...

- P1L13: The statement that the GW forcing is the reason for all of the changes here is too absolute, at least you need to add a 'mainly', because planetary waves also have a significant contribution here. But see my points below that explain the topic.

- P1L13-14: ... at the equatorward upper flank of the subtropical jet.

- P1L17: ... with observed and modeled BDC changes.

- P2L28: ozone depleting substances

- P3L4: ... consistent with observed negative ...

- P3L30: ...includes extensive improvements...

- P4L14: exchange order of '80km' and '31km', for consistency

- P4L18: Can you be more specific than 'higher up'?

- P4L19: Through the higher spatial and.... are a better..., a better .... . Moreover, data from many recent satellite instruments are now additionally assimilated.

- P4L29: remove 'h'

- P7L21: ...normalized coefficients the QBO and ....

- P8L10: three distinct regions of the stratosphere (tropical pipe, mid-latitude surf zone and polar regions)

- P8L19: remove 'at 95% confidence interval'

- P8L31: To quantify the circulation differences

- P8L33: The vertical w* profiles

- P8L34: in the w* structure

- P9 caption Fig.1: Write more compact: '...annual (a-c), DJF (d-f) and JJA (g-i) mean ...' and remove the two sentences below (the same applies to caption of Fig. 4 and Fig. 5)

- P9 caption Fig.1: ... . Grey line indicates the zero $\overline{w^*}$ contours. Grey dots ...

- P10 Caption Fig. 2: ... (b) tropical ...

- Fig. 2: Change header of figures: Tropical upwelling is a mass flux, but here you show w*, the residual vertical velocity.

- Fig. 2: State in the captions or text what latitude bands you use, or if you use the turnaround latitudes to determine the regions of up- and downwelling (or did I miss that?).

- P10L1: the relative $\overline{w^*}$ differences

- P10L2 the large-scale downwelling differences

- P11 caption Fig.3: remove 'together with'

- P11 caption Fig.3: horizontal lines indicate

- P11 caption Fig.3: climatologies. (remove 'from the ERA5 and ERA-Interim reanalyses respectively')

- P11L5 Change 'errorbars' to 'the uncertainties' or to 'variability is'

- P11L15 of the $\overline{w^*}$ differences between

- P12L7: remove 'remarkably'

- P12L11-12: stream-function upwelling in the tropical pipe and the mid-latitude surf zone is weaker in ERA5 than in ...

- P15L1 with the $\overline{w^*}$ differences

- P15L12: remove 'remarkably'

- P15L25: 'increase with increasing' sounds odd. Maybe: 'increase with enhanced'

- P15L16f: Rephrase sentence to: 'Therefore, the use of...drag parameterization in ERA5 likely is the cause of...'

- P15L30: Rephrase sentence to: 'In ERA-Interim, Rayleigh drag was applied as a substitute...'

- P15L30f: Rephrase sentence to: 'For ERA5, a Warner and McIntyre (2001) type non-orographic spectral gravity wave scheme was introduced and hence the Rayleigh drag could be switched off.'

- P16L3: ... agree well in phase ...

- P16L11: ... (Fig. 7c, d). Westerly shear reduces....

- P17L15: one of the major

- P17L23: ...in $\overline{w^*}$ and $\psi^*$, the regression analysis...

- P17L28: ...in $\overline{w^*}$ show...

- P17L29: ...in $\overline{w^*}$ show...

- P19 caption Fig.8: Remove 'horizontal'

- P19 Fig.8: Add overbars to w*, also in other figures, and also to $\psi^*$ in captions and figures.

- P20L5: Chage 'therefore' to 'thereby'

- P21L4-5: ...reveals larger negative $\overline{w^*}$ anomalies than ERA-Interim, which is likely due to the differences in wave activity(...).

- P21L11: Change 'This' to 'These

- P21L30 and L31 (two times): remove 'would' and change 'weaken' to 'weakens'

- P23L5 while the ... upper stratosphere. I din't think that sentence make sense as it is, better revise it.

- P23L6 remove 'previously reported'

- P23L8 "difference is less evident'. But there are still significant differences in the lower stratosphere, that should be mentioned. Moreover, there are strong PW differences in the upper troposphere, but I know these are not meant here (still they could be mentioned, and stated that they are not meant for this or that reason). However, the text must therefore be cautiously revised as to whether the upper troposphere, or the lower stratosphere is meant. Here, you state UTLS and that is not correct. Also in Line 11 you incorrectly state UTLS, please go through the entire text to make it more precise.

- P23L13: ... two reanalyses are governed by differences in the contribution of both the planetary and...

- P23L19 ...based on the non-orographic gravity wave parameterization...

- P23L20 remove the sentence "This means that... any longer" (that is clear)

- P23L26 and L35: the gravity wave breaking differences are mostly in the lower stratosphere. There are also clear differences in planetry wave forcing, the strongest in the upper tropopshere, but also some in the lower stratosphere. Please take that into account here, and add it to the text and be precise about the regions, as written above.

- P23L30 ...in both hemispheres, but can be seen in a much larger ...

- P24 Figure 11: I think you must (additionally) show the differences here, because I cannot everywhere see what you state. I see it in the SH, but in the NH, the differences are not large enough to see the differences by eye. Moreover, this will help you to quantify the contributions of GWs and of planetary waves and then you can in the text more clearly discuss how much which contribution is.

- P24 Figure 11: You need to explain why you chose 70 hPa. It is easily explainable citing that 70 hPa is traditionally (citing Butchart et al. (2010), Hardiman et al. (2014)...) used, and Dietmüller et al. 2018 (https://doi.org/10.5194/acp-18-6699-2018) show that 70 hPa works best for AoA and for RCTT in an inter-model correlation for almost all the stratosphere. Moreover, as all your other figures show geometric altitude, for reference, state what 70 hPa more or less refers to in km altitude.

- P24L3: correct UTLS as mentioned above

- P25L1-2 I think this sentence is somewhat too general, please specifiy it at least concerning the altitude and moreover, state that there are also significant differences in PW driving that contribute there, although somewhat smaller than the gravity wave changes.

- P25L13: 'boundary layer' is not a process, maybe write boundary layer physics if that is what is meant.

- P25L20 change 'projections' to 'simulations'

- P25L22 remove 'at 95%'

- P25L30 I think you want to refer to panel g in Fig. 10 here. The patterns here also indicate that the effect is stonger in the SH than in the NH, which goes together with the BDC differences

- P25L31 "MAINLY by enhanced gravity wave breaking"

- P26 figure 12: Mention in the caption that the differences are not significant on the 95% level anywhere here if that is correct. If not, include the dots.

- P26L1: Change 'Eichinger and Sacha 2020' to 'Sacha et al 2019', that fits better

- P26L6: ... using stratospheric age of air and its spectrum...

- P27L9: Here you can add the Eichinger and Sacha 2020 citation

- P27L10-11: remove: 'at 95% using student t-test with two tail distribution'

- P27L18 and 19: Provide the std declaration behind the number in brackets with unit. I.e. that way: ($\sigma = 0.053\ kg \cdot s^{-1} \cdot dec^{-1}$)

- P27L20 ... but it is significant and between 10% and 20% below 70 hPa. This indicates that....

- P27L27 In our comparisons...

- P27L27 Remove 'a remarkably'

- P27L31-32 I guess you are only talking about the deep branch (or the downwelling regions) here, include that to the sentence, otherwise it is very confusing

- P27L33-34: Again, that statement is much to absolute fo me. At least you must add a 'mostly' or alike in front of the gravity wave, because planetary waves also contribute significantly.

- P28 caption Fig. 13: A trend is a scalar, while the lines you show here are functions. Therefore, change 'trend' to 'linear regression lines' (twice).

- P28L13: change 'a very' to 'show'

- P29L9: I think 'less evident' is not a good way to put it, as the differences are after all significant. But with the difference of the downward control analysis you can make this statement more quantitative.

- P29L11 larger than what? You forgot to clearly state what reanalysis is larger here, and in the following which BDC in the UTLS is stronger. Moreover, as discussed above, UTLS is not the term you want to use here. Revise also this complete section w.r.t. that inaccuracy.

- P29L12: Therefore, these differences ...

- P29L14: ...scheme. Moreover, progress has been made....

- P29L17: Change 'Even not' to 'Although not'

- P29L17f remove 'with the student t-test'

---

## Author Response (AR2)

**Answer to Roland Eichinger's comments on "The advective Brewer-Dobson circulation in the ERA5 reanalysis: climatology, variability and trends" by Mohamadou Diallo et al.**

Dear Editor-in-Chief, Peter H. Haynes,

We are submitting our revised article titled "'The advective Brewer-Dobson circulation in the ERA5 reanalysis: climatology, variability and trends". We thank the Reviewer #1 for already accepting the manuscript as it is and the Reviewer #2 for detailed and well thought-out minor comments, which helped to further improve the paper. We have made substantial changes to the manuscript in order to thoroughly address the Reviewers' suggestions and comments. Main changes concern:

- We have substituted the panels (c, d) in the fig 11 by the differences between ERA5 and ERAi as suggested by Reviewer #1, a new figure showing these results and the related discussion.

- rephrasing of several paragraphs in order to clarify the manuscript.

With these changes, we are convinced that the paper has been significantly improved and is highly relevant for a wide-ranging journal like *Atmospheric Chemistry and Physics*. Please see below our answers point by point to all reviewers comments and suggestions.

Reviewers comments are in bold, followed by our respective replies. Changes in the manuscript are in blue, allowing them to be tracked easily.

Kind regards,
Mohamadou Diallo (on behalf of the co-authors)

**Roland Eichinger, Reviewer #2 (Comments to Author):**

**Minor issues:**

1. **P1L4: ...inter-annual variability, climatology and trends...**

   We have rephrase it.

2. **P1L4: ...with the predecessor ERA-Interim...**

   We have rephrased the sentence.

3. **P1L13: The statement that the GW forcing is the reason for all of the changes here is too absolute, at least you need to add a mainly, because planetary waves also have a significant contribution here. But see my points below that explain the topic.**

   The contribution of planetary wave is significant but weaker than the gravity wave as clearly shown by the differences in Fig. 11. Most importantly, the contribution of planetary wave occurs either below the tropopause or far from the regions known as important for wave driving the upwelling branch of the BDC. We have clarified it.

4. **P1L13-14: ... at the equatorward upper flank of the subtropical jet.**

   We have rephrased it.

5. **P1L17: ... with observed and modeled BDC changes.**

   We have rephrased it.

6. **P2L28: ozone depleting substances**

   We have rephrased it.

7. **P3L4: ... consistent with observed negative ...**

   We have rephrased it.

8. **P3L30: ...includes extensive improvements...**

   We have rephrased it.

9. *P4L14: exchange order of "80km" and "31km", for consistency*

   We have rephrased it.

10. *P4L18: Can you be more specific than "higher up"?*

    We have rephrased it.

11. *P4L19: Through the higher spatial and.... are a better..., a better .... . Moreover, data from many recent satellite instruments are now additionally assimilated.*

    We have rephrased it.

12. *P4L29: remove "h"*

    We have rephrased it.

13. *P7L21: ...normalized coefficients the QBO and ....*

    We have rephrase it.

14. *P8L10: three distinct regions of the stratosphere (tropical pipe, mid-latitude surf zone and polar regions*

    We have rephrased it.

15. *P8L19: remove "at 95% confidence interval"*

    We have removed it.

16. *P8L31: To quantify the circulation differences*

    We have rephrased it.

17. *P8L33: The vertical w\* profiles*

    We have rephrased it.

18. *P8L34: in the w\* structure*

    We have rephrased it.

19. *P9 caption Fig.1: Write more compact: "...annual (a-c), DJF (d-f) and JJA (g-i) mean ..." and remove the two sentences below (the same applies to caption of Fig. 4 and Fig. 5)*

    We have rephrased it.

20. *P9 caption Fig.1: ... . Grey line indicates the zero w\* contours. Grey dots ...*

    We have rephrased it.

21. *P10 Caption Fig. 2: ... (b) tropical ...*

    We have rephrased it.

22. *Fig. 2: Change header of figures: Tropical upwelling is a mass flux, but here you show w\*, the residual vertical velocity.*

    We decided to keep the wording as is.

23. *Fig. 2: State in the captions or text what latitude bands you use, or if you use the turnaround latitudes to determine the regions of up- and downwelling (or did I miss that?).*

    This is already stated clearly in the text (Page 9, line 28-30). We have also added it in the caption.

24. *P10L1: the relative w\* differences*

    We have rephrased it.

25. *P10L2 the large-scale downwelling differences*

    We have rephrased it.

26. *P11 caption Fig.3: remove "together with"*

    We have rephrased the sentence.

27. **P11 caption Fig.3: horizontal lines indicate.**

    We have rephrased it.

28. **P11 caption Fig.3: climatologies. (remove "from the ERA5 and ERA-Interim reanalyses respectively")**

    We have removed it.

29. **P11L5 Change "errorbars" to "the uncertainties" or to "variability is"**

    We have rephrased it.

30. **P11L15 of the w\* differences between**

    We have rephrased it.

31. **P12L7: remove "remarkably"**

    We have removed it.

32. **P12L11-12: stream-function upwelling in the tropical pipe and the mid-latitude surf zone is weaker in ERA5 than in ...**

    We have rephrased it.

33. **P15L1 with the w\* differences**

    We have rephrased it.

34. **P15L12: remove "remarkably"**

    We have remove it.

35. **P15L25: "increase with increasing: sounds odd. Maybe: "increase with enhanced"**

    We have rephrased it.

36. **P15L16f: Rephrase sentence to: "Therefore, the use of...drag parameterization in ERA5 likely is the cause of..."**

    We have rephrased it.

37. **P15L30: Rephrase sentence to: "In ERA-Interim, Rayleigh drag was applied as a substitute..."**

    We have rephrased it.

38. **P15L30f: Rephrase sentence to: "For ERA5, a Warner and McIntyre (2001) type nonorographic spectral gravity wave scheme was introduced and hence the Rayleigh drag could be switched off."**

    We have rephrased it.

39. **P16L3: ... agree well in phase ...**

    We have rephrased it.

40. **P16L11: ... (Fig. 7c, d). Westerly shear reduces....**

    We have rephrased it.

41. **P17L15: one of the major**

    We have rephrased it.

42. **P17L23: ...in w\* and $\psi$\*, the regression analysis...**

    We have rephrased it.

43. **P17L28: ...in w\* show...**

    We have rephrased it.

44. **P17L29: ...in w\* show...**

    We have rephrased it.

45. ***P19 caption Fig.8: Remove "horizontal"***

   We have rephrased it.

46. ***P19 Fig.8: Add overbars to w*, also in other figures, and also to ψ* in captions and figures.***

   With the matlab version that we have, it is not possible to put the overbar in the figures. We have decided to keep it as it is.

47. ***P20L5: Change "therefore" to "thereby"***

   We have rephrased it.

48. ***P21L4-5: ...reveals larger negative w* anomalies than ERA-Interim, which is likely due to the differences in wave activity(...).***

   We have rephrased it.

49. ***P21L11: Change "This" to "These"***

   We have rephrased it.

50. ***P21L30 and L31 (two times): remove "would" and change "weaken" to "weakens"***

   We have rephrased it.

51. ***P23L5 while the ... upper stratosphere. I dint think that sentence make sense as it is, better revise it.***

   We have rephrased it.

52. ***P23L6 remove "previously reported"***

   We have rephrased it.

53. ***P23L8 "difference is less evident". But there are still significant differences in the lower strato-sphere, that should be mentioned. Moreover, there are strong PW differences in the upper tropo-sphere, but I know these are not meant here (still they could be mentioned, and stated that they are not meant for this or that reason). However, the text must therefore be cautiously revised as to whether the upper troposphere, or the lower stratosphere is meant. Here, you state UTLS and that is not correct. Also in Line 11 you incorrectly state UTLS, please go through the entire text to make it more precise.***

   Please see item 3 above. We have rephrased the text accordingly.

54. ***P23L13: ... two reanalyses are governed by differences in the contribution of both the planetary and...***

   We have rephrased it.

55. ***P23L19 ...based on the non-orographic gravity wave parameterization...***

   We have rephrased it.

56. ***P23L20 remove the sentence "This means that... any longer" (that is clear)***

   We have removed it.

57. ***P23L26 and L35: the gravity wave breaking differences are mostly in the lower stratosphere. There are also clear differences in planetry wave forcing, the strongest in the upper tropopshere, but also some in the lower stratosphere. Please take that into account here, and add it to the text and be precise about the regions, as written above.***

   Thank for this comments. note that wave breaking near the equatorward upper flank of the subtropical jet are the ones that initially drive the upwelling branch of the BDC. In this region GW contribution is the dominant one and the area of significant is even much larger than the planetary wave. The planetary wave contribution are significant but weaker than compared to the gravity wave contribution near the upper flank of the subtropical jet. In addition, the PW contribution as you notice is below the tropopause or outside of this key region. We have rephrased it and clarified it.

58. ***P23L30 ...in both hemispheres, but can be seen in a much larger ...***

   We have removed it.

59. ***P24 Figure 11: I think you must (additionally) show the differences here, because I cannot everywhere see what you state. I see it in the SH, but in the NH, the differences are not large enough to see the differences by eye. Moreover, this will help you to quantify the contributions of GWs and of planetary waves and then you can in the text more clearly discuss how much which contribution is.***

We have changed the Fig 11 by substituting the ERA-interim panel by the different between ERA5 and ERA-Interim. The text has been modified accordingly.

60. ***P24 Figure 11: You need to explain why you chose 70 hPa. It is easily explainable citing that 70 hPa is traditionally (citing Butchart et al. (2010), Hardiman et al. (2014)...) used, and Dietmuller et al. 2018 (https://doi.org/10.5194/acp-18-6699-2018) show that 70 hPa works best for AoA and for RCTT in an inter-model correlation for almost all the stratosphere. Moreover, as all your other figures show geometric altitude, for reference, state what 70 hPa more or less refers to in km altitude.***

Thanks for the suggestion. Actually, we had already justified the use of 70hPa for the figure 2 (Page 11, Line 3-5) and all these references were already cited in the manuscript. We have added also at Page 23, Line 33-35.

61. ***P24L3: correct UTLS as mentioned above***

We have rephrased it.

62. ***P25L1-2 I think this sentence is somewhat too general, please specifiy it at least concerning the altitude and moreover, state that there are also significant differences in PW driving that contribute there, although somewhat smaller than the gravity wave changes.***

We have removed it.

63. ***P25L13: "boundary layer" is not a process, maybe write boundary layer physics if that is what is meant.***

We have removed it.

64. ***P25L20 change "projections" to "simulations"***

We have removed it.

65. ***P25L22 remove "at 95%"***

We have removed it.

66. ***P25L30 I think you want to refer to panel g in Fig. 10 here. The patterns here also indicate that the effect is stonger in the SH than in the NH, which goes together with the BDC differences.***

The figure reference was not correct. It was fig. 12 here. We have changed it.

67. ***P25L31 "MAINLY by enhanced gravity wave breaking"***

We have removed it.

68. ***P26 figure 12: Mention in the caption that the differences are not significant on the 95% level anywhere here if that is correct. If not, include the dots.***

We have mentioned that time in the text and also added to the caption.

69. ***P26L1: Change "Eichinger and Sacha 2020" to "Sacha et al 2019", that fits better***

We have rephrased it.

70. ***P26L6: ... using stratospheric age of air and its spectrum...***

We have rephrased it.

71. ***P27L9: Here you can add the Eichinger and Sacha 2020 citation***

We have rephrased it.

72. ***P27L10-11: remove: "at 95% using student t-test with two tail distribution"***

We decided to keep it as it is because it's good to remind it in the conclusion.

73. ***P27L18 and 19: Provide the std declaration behind the number in brackets with unit. I.e. that way: (σ = 0 : 053kg/s/dec)***

We have changed by adding the sigma directly to the trend as plus/minus uncertainty.

74. ***P27L20 ... but it is significant and between 10% and 20% below 70 hPa. This indicates that....***

We decided to keep it as it is.

75. ***P27L27 In our comparisons...***

We have rephrased it.

76. ***P27L27 Remove "a remarkably"***

We decided to keep it as it is because the structure of the circulation agreed remarkably well actually.

77. ***P27L31-32 I guess you are only talking about the deep branch (or the downwelling regions) here, include that to the sentence, otherwise it is very confusing***

We are talking about the overall BDC and then region region by region. We decided to keep it as it is.

78. ***P27L33-34: Again, that statement is much to absolute for me. At least you must add a mostly or alike in front of the gravity wave, because planetary waves also contribute significantly.***

We have rephrased it.

79. ***P28 caption Fig. 13: A trend is a scalar, while the lines you show here are functions. Therefore, change "trend" to "linear regression lines" (twice).***

For consistency with the discussion in our regression section, and to be more accurate, we replaced "trend" with "linear trend".

80. ***P28L13: change "a very" to "show"***

We have added the missing verb.

81. ***P29L9: I think "less evident" is not a good way to put it, as the differences are after all significant. But with the difference of the downward control analysis you can make this statement more quantitative.***

We have rephrased it accordingly to the Fig. 11 which confirm that the GW contribution dominates.

82. ***P29L11 larger than what? You forgot to clearly state what reanalysis is larger here, and in the following which BDC in the UTLS is stronger. Moreover, as discussed above, UTLS is not the term you want to use here. Revise also this complete section w.r.t. that inaccuracy.***

We have rephrased it.

83. ***P29L12: Therefore, these differences ...***

We have rephrased it.

84. ***P29L17: Change "Even not" to "Although not"***

We have rephrased it.

85. ***P29L17f remove "with the student t-test"***

We have rephrased it.